# Immune suppressive activity of myeloid-derived suppressor cells in cancer requires inactivation of the type I interferon pathway

Kevin Alicea-Torres[1,8], Emilio Sanseviero[1,2,8], Jun Gui[3,6], Jinyun Chen[3], Filippo Veglia[1,7], Qiujin Yu[3], Laxminarasimha Donthireddy[1], Andrew Kossenkov[1], Cindy Lin[1], Shuyu Fu[1], Charles Mulligan [4], Brian Nam[4], Gregory Masters[4], Fred Denstman[4], Joseph Bennett[4], Neil Hockstein[4], Agnieszka Rynda-Apple[5], Yulia Nefedova[1], Serge Y. Fuchs [3] & Dmitry I. Gabrilovich [2✉]

Myeloid-derived suppressor cells (MDSC) are pathologically activated neutrophils and monocytes with potent immune suppressive activity. These cells play an important role in accelerating tumor progression and undermining the efficacy of anti-cancer therapies. The natural mechanisms limiting MDSC activity are not well understood. Here, we present evidence that type I interferons (IFN1) receptor signaling serves as a universal mechanism that restricts acquisition of suppressive activity by these cells. Downregulation of the IFNAR1 chain of this receptor is found in MDSC from cancer patients and mouse tumor models. The decrease in IFNAR1 depends on the activation of the p38 protein kinase and is required for activation of the immune suppressive phenotype. Whereas deletion of IFNAR1 is not sufficient to convert neutrophils and monocytes to MDSC, genetic stabilization of IFNAR1 in tumor bearing mice undermines suppressive activity of MDSC and has potent antitumor effect. Stabilizing IFNAR1 using inhibitor of p38 combined with the interferon induction therapy elicits a robust anti-tumor effect. Thus, negative regulatory mechanisms of MDSC function can be exploited therapeutically.

[1] The Wistar Institute, Philadelphia, PA, USA. [2] AstraZeneca, Gaithersburg, MD, USA. [3] Department of Biomedical Sciences, School of Veterinary Medicine University of Pennsylvania, Philadelphia, PA, USA. [4] Helen F. Graham Cancer Center and Research Institute, Newark, DE, USA. [5] Department of Microbiology and Immunology, Montana State University, Bozeman, MT, USA. [6] Present address: State Key Laboratory of Oncogenes and Related Genes, Stem Cell Research Center, Renji Hospital, School of Medicine Shanghai Jiao Tong University, Shanghai, China. [7] Present address: H. Lee Moffitt Cancer Center, Tampa, FL, USA. [8] These authors contributed equally: Kevin Alicea-Torres, Emilio Sanseviero. ✉email: dmitry.gabrilovich@astrazeneca.com

Myeloid-derived suppressor cells (MDSC) are pathologically activated neutrophils (PMN) and monocytes (Mon) with potent immune-suppressive activity. These cells have a distinct transcriptional profile, biochemical features and, most importantly, exhibit potent ability to suppress immune responses[1]. The total population of MDSC consists of three groups of cells: the most abundant (>75%) relatively immature, PMN-MDSC; the less abundant population of pathologically activated monocytes—(M-MDSC); and a small population of early myeloid precursors present in humans[2]. MDSC accumulation is described in many pathologic conditions, but they play an especially prominent role in cancer as major regulators of immune responses and one of the limiting factors for cancer immunotherapy. In recent years, the clinical role of MDSC has emerged. Results showed a positive correlation of MDSC in peripheral blood (PB) with cancer stage and tumor burden in many types of cancer[3–5]. Elevated MDSC in the circulation was found to be an independent indicator of poor outcomes in patients with solid tumors[6]. Recent studies demonstrated values of MDSC in predicting response to therapy in many types of cancer[7–13]. The circulating MDSC negatively correlated with objective clinical response to check-point inhibitors[14–17]. In tumor mouse models, MDSC inhibition during immunotherapy increases its therapeutic effect[18–23].

The current paradigm stipulates that PMN and Mon acquire immune suppressive features during their differentiation as a result of the direct effect of various tumor-derived factors and tumor microenvironment. Several cytokines and growth factors, as well as a number of transcriptional factors and signal transduction pathways, were implicated in the promotion of immune suppressive function of MDSC[24]. However, recent studies demonstrated that only a fraction of PMN (usually around 5–10% in PB and 60–70% in tumor tissues) in cancer patients acquire features of MDSC[25]. The fact that the proportion of MDSC in cancer patients correlated with the clinical outcome also indicates that MDSC represents only part of populations of PMN and Mon. This raises an important question: why do tumor-derived factors abundantly present in tumor-bearing (TB) hosts not affect all cells? We hypothesized that in addition to "positive" signals that promote MDSC, there are also "negative" signals that limit the acquisition of MDSC features by PMN and Mon. Furthermore, these negative regulatory mechanisms are expected to be somehow inactivated in myeloid cells exposed to the stimuli derived from malignant cells or/and present in the tumor microenvironment.

During the analysis of previously obtained MDSC gene expression profile data, we noticed a substantial downregulation of the type I interferons (IFN1) pathway[26]. IFN1 (including IFN-α and IFN-β) act on cells by engaging a cognate receptor (consisting of two chains, IFNAR1 and IFNAR2) and triggering the signal transduction pathway that involves activation of TYK2/JAK1 Janus kinases, phosphorylation, and activation of STAT1/STAT2 heterodimer. The latter interacts with IRF9, translocates to the nucleus, and trans-activates the IFN1-stimulated genes[27–29]. IFN1 is known to promote anti-tumor immunity by diverse mechanisms including the stimulation of DC maturation and antigen presentation and improving the survival of cytotoxic lymphocytes[30–33]. IFN1 is suggested to enhance the effect of cancer immunotherapy[28,34]. Tumor-derived factors (such as pro-inflammatory cytokines[35] or extracellular vesicles[36] and stimuli of the tumor microenvironment (deficit of oxygen or amino acids[37]) inactivate the IFN1 pathway via prompting the p38 kinase-dependent phosphorylation, ubiquitination, and degradation of IFNAR1[33,38,39]. However, the role of IFN1 in the regulation of numbers and suppressive activities of MDSC remains poorly understood.

Furthermore, the literature on the role of IFN1 in the control of MDSC is rather controversial. Reports suggesting that IFN1 stimulate the generation[40] or suppressive activities of MDSC via sustaining expression of PD-L1[41] are contradicted by the studies where administration of IFN1 inducers such as poly(I:C) or CpG led to decreased numbers or/and undermined suppressive activities of MDSC[42–44]. Whereas activation of STING and ensuing production of IFN1 by tumor irradiation was suggested to induce MDSC[45], the opposite results were reported by studies using the forced expression of STING; however, the latter effects were not dependent on IFN1[46]. Thus, the critical questions regarding the possible role of IFN1 in the regulation of MDSC suppressive activity as well as the mechanisms by which the effects of IFN1 on MDSC are inactivated in the tumor microenvironment remain to be answered.

In this work, we show that mouse and human MDSC exhibit a substantially reduced expression of IFNAR1 on their surface compared to PMN and Mon. This results in inhibition of the IFN1 pathway. Loss of IFNAR1 alone is not sufficient to convert PMN or Mon to MDSC; however, preventions of IFNAR1 degradation abrogates MDSC activity and have a potent anti-tumor effect.

## Results

**MDSC in cancer patients and TB mice exhibit low expression of type I interferon receptor.** Careful analysis of the previously obtained transcriptome of mouse spleen PMN and PMN-MDSC[26] revealed a substantial downregulation of the IFNAR1 mediated pathway in PMN-MDSC (Supplementary Fig. 1A). We found 911 genes that were significantly downregulated (FDR < 20%) in PMN-MDSC compared to PMN. Among these genes, there was a significant enrichment of interferon-induced genes (2.4-fold over random chance, $p = 7 \times 10^{-6}$ by Fisher exact test). Similar observations were made during the analysis of the transcriptome of blood PMN and PMN-MDSC (based on LOX-1 expression) in cancer patients[25] (Supplementary Fig. 1B). These results suggested that the interferon pathways are suppressed in the PMN-MDSC in cancer patients and this phenotype is recapitulated in the PMN-MDSC from the TB mice.

All responses to IFN1 are determined by the cell surface levels of the IFN1 receptor, which, in turn, is tightly controlled by ubiquitination and degradation of its IFNAR1 chain[39,47–49]. We compared IFNAR1 expression on the surface of PMN and Mon from healthy donors and patients with non-small cell lung cancer, pancreatic cancer, breast cancer, head and neck cancer, and colon cancer (Supplementary Fig. 2A). Although several patients demonstrated decreased expression of IFNAR1 in Mon as compared to healthy donors, in the entire group, the differences were not statistically significant (Fig. 1a). In contrast, cancer patients' PMN-MDSC demonstrated markedly lower expression of IFNAR1 than PMN from healthy donors (Fig. 1a). Those differences in IFNAR1 expression were largely associated with the presence of MDSC in cancer patients. When Mon and M-MDSC, as well as PMN and PMN-MDSC, were compared in the same patients separated using established phenotypic criteria[2] (Supplementary Fig. 2B), substantially lower amounts of the IFNAR1 on M-MDSC and PMN-MDSC than Mon and PMN were observed (Fig. 1b). Notably, the decrease in IFNAR1 expression was more pronounced in PMN-MDSC than in M-MDSC.

A similar analysis was performed on mouse MDSC, which were gated using established criteria: PMN-MDSC—CD11b[+]Ly6C[lo]Ly6G[+], M-MDSC—CD11b[+]Ly6C[hi]Ly6G[−2]. Cells with the same phenotype in naïve, tumor-free mice were defined as PMN and Mon, respectively (Supplementary Fig. 2C). We assessed the expression of IFNAR1 in MDSC from transplantable

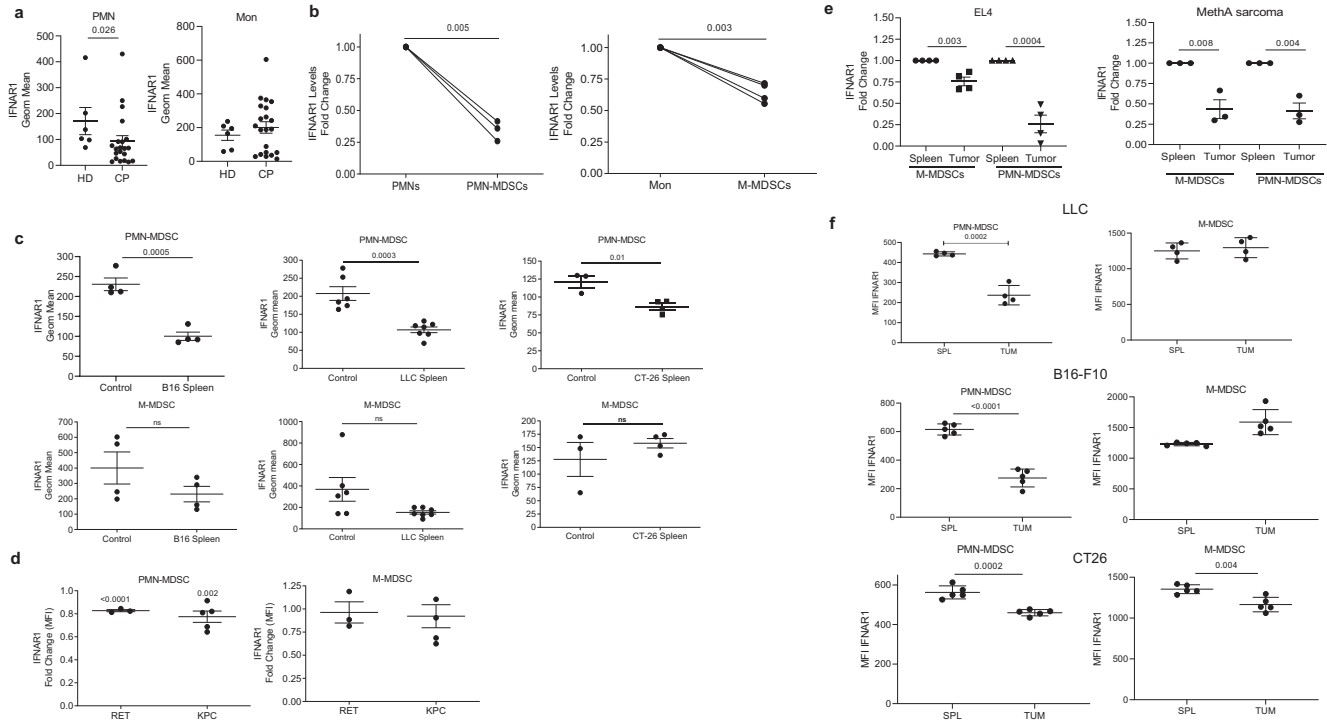

**Fig. 1 IFNAR1 is downregulated in MDSCs. a** IFNAR1 levels on CD15+ PMN (left) and CD14+ Mon (right) from whole blood of healthy donors (HD) or cancer patients (CP). Individual results in 6 healthy donors and 22 cancer patients are shown. Mean ± SD of IFNAR1 expression is presented. P values were calculated in Mann–Whitney test. **b** IFNAR1 measured on PMN versus PMN-MDSCs (n = 3) or Mon versus M-MDSCs (n = 4) from the same patients. Fold change from IFNAR1 level in PMN or Mon in each patient is shown. **c** IFNAR1 levels on mouse splenic PMN-MDSCs and M-MDSCs from indicated s.c. tumor models analyzed by flow cytometry. Each dot represents the geometric mean of IFNAR1 levels in individual mice (n = 4 for B16 model, n = 7 for LLC mode, n = 4 for CT26 model). **d** IFNAR1 levels on splenic PMN-MDSCs and M-MDSCs from spontaneous tumor models analyzed by flow cytometry. Fold change over spleen of tumor-free mice (n = 3 for RET model, n = 5 for KPC model PMN-MDSC and n = 4 for M-MDSC). Mean and SD is shown. **e** IFNAR1 levels on PMN-MDSCs and M-MDSCs from spleen and indicated ascites tumors analyzed by flow cytometry. Each dot represents an individual mouse (n = 4 for EL4 model, n = 3 for MethA model). IFNAR1 levels is presented as fold changes from spleen cells. **f** IFNAR1 levels on PMN-MDSCs and M-MDSCs from spleen and mechanically dissociated tumors analyzed by flow cytometry. Each dot represents the geometric mean of IFNAR1 levels in individual mice. (n = 5 for B16-F10 and CT26 models and n = 4 for LLC model). In all panels, data are expressed as mean ± SD and the p values were calculated using unpaired two-sided Student's t test. ns—p > 0.05.

models of B16 melanoma, LLC lung carcinoma, CT26 colon carcinoma (Fig. 1c), and genetically engineered models of melanoma (RET) and pancreatic cancer (KPC) (Fig. 1d). In transplantable models, spleen PMN-MDSC demonstrated markedly lower expression of IFNAR1 than corresponding PMN from tumor-free mice. Although in some models there was a clear trend in the decrease of IFNAR1 expression in M-MDSC as compared to Mon, it did not reach statistical significance (Fig. 1c). In RET and KPC models, IFNAR1 levels were significantly lower in PMN-MDSC compared with PMN, whereas M-MDSC and Mon exhibited comparable same levels of the receptor (Fig. 1d). To safeguard against potential IFNAR1 cleavage by proteases[50], we avoided enzymatic tumor digestion by using ascites of EL4 and MethA sarcoma tumors where myeloid cells can be directly isolated from the tumor site. We observed dramatically reduced expression of IFNAR1 on the surface of tumor PMN-MDSC and M-MDSC than on spleen cells (Fig. 1e). We also used mechanical disruption of tumors without enzymatic digestion to recover PMN-MDSC and M-MDSC. IFNAR1 expression in tumor PMN-MDSC was substantially lower than in spleen PMN-MDSC, whereas only in CT26 TB mice it was lower in M-MDSC (Fig. 1f). Downregulation of IFNAR1 in PMN-MDSC and M-MDSC was associated with dramatically lower expression of the IFN1-inducible genes such as *Irf7* and *Isg15* (Fig. 2a and Supplementary Fig. 3A). Thus, both populations of MDSC in the blood of cancer patients, as well as in tumors of TB mice and PMN-MDSC in

spleens of TB mice exhibited marked inhibition of the IFN1-IFNAR1 pathway.

**The biological role of IFNAR1 in MDSC**. Given the difference in IFNAR1 levels in tumor and spleen MDSCs, we next compared side-by-side functional activities of MDSC in tumors and spleens. Tumor PMN-MDSC and M-MDSC were markedly more suppressive than spleen MDSC (Fig. 2b). This suggested a possible association between the functional activity of MDSC and the expression of IFNAR1. We asked whether the suppressive activity of MDSC could be canceled by providing high amounts of IFNAR1 ligand—IFNβ. PMN-MDSC and M-MDSC isolated from the spleen of TB mice were cultured for 2 h with IFNβ (2000U), extensively washed, and then used in a suppressive assay. This treatment completely abrogated PMN-MDSC and M-MDSC suppressive activity (Fig. 2c) indicating that IFN1 acts as a negative regulator of MDSC immune suppressive function.

Next, we investigated whether loss of IFNAR1 signaling was sufficient to convert PMN or Mon to MDSC by using *Ifnar1* KO mice. PMN or Mon from tumor-free *Ifnar1* KO mice indeed lacked IFNAR1 (Supplementary Fig. 3B) and yet did not display suppressive activity (Fig. 2d). While antiviral effects of IFN1 can occur at low levels of the IFN1 receptor, even a modest downregulation of IFNAR1 notably diminishes its immunopathologic activities[39,48]. Thus, we hypothesized that downregulation of

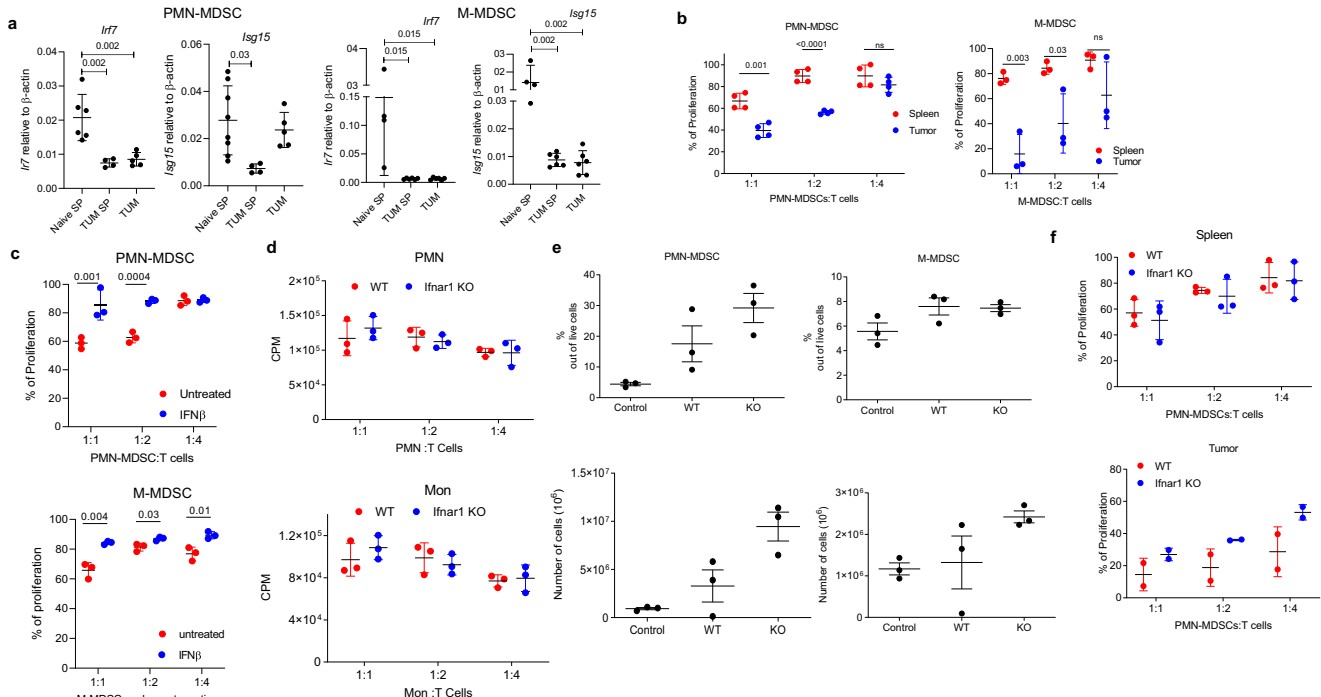

**Fig. 2 Lack of IFNAR1 does not alter immune-suppressive activity by PMN-MDSCs. a** Expression of interferon response genes in MDSC. PMN and monocytes were isolated by cell sorting from the spleen of naïve mice (Naïve SP) or spleen (TUM SP) and tumors (TUM) of EL4-bearing mice. *Irf7* and *Isg15* expression were quantified by qRT-PCR in each cell population. Each dot represents a single mouse. Mean and SD is shown. In PMN-MDSC: $n = 6$ for Naïve SP *Irf7*, $n = 4$ for TUM SP *Irf7*, $n = 5$ for TUM *Irf7*, $n = 8$ for Naïve SP *Isg15*, $n = 4$ for TUM SP, $n = 5$ for TUM *Isg15*. In M-MDSC: $n = 4$ for Naïve SP *Irf7* group, $n = 6$ for TUM SP and TUM groups, $n = 4$ for Naïve SP *Isg15*, $n = 6$ for TUM SP *Isg15*, $n = 6$ for TUM *Isg15*. *P* values were calculated in the ANOVA test with corrections for multiple comparisons. **b** Suppression assay of PMN-MDSCs and M-MDSCs isolated from both spleen and tumors of tumor-bearing mice. PMN-MDSCs ($n = 4$) and M-MDSCs ($n = 3$) from spleen or tumor were cocultured with antigen-specific CD8⁺ T cells (OT-1 splenocytes) at different ratios. T cell proliferation was evaluated in triplicate using [³H]-thymidine uptake and presented as a percentage based on positive control (without MDSCs), set as 100% proliferation. **c** Suppression assay (as in **a**) of splenic PMN-MDSC and M-MDSC treated for 2 h with 2000 U of IFNβ. ($n = 3$). Mean and SD is shown. **d** PMN and Mon were isolated from bone marrow (BM) of tumor-free WT or *Ifnar1* KO mice. Suppression assay was performed as described in (**b**). **e** Percentage and an absolute number of PMN-MDSCs and M-MDSCs in the spleen of tumor-bearing mice. Mean and SD is shown. $n = 3$. **f** Suppressive activity of PMN-MDSCs from spleen and tumors. $N = 3$ for spleen, $n = 2$ for tumor. A representative of three experiments. 100%—proliferation in the absence of MDSC. In all panels, data are expressed as mean ± SD and the *p* values were calculated using unpaired two-sided Student's *t* test, except when otherwise indicated. \**p* < 0.05; \*\**p* < 0.01; \*\*\**p* < 0.001; \*\*\*\**p* < 0.0001. ns not significant.

IFNAR1 protein in TB mice could phenocopy genetic ablation of the *Ifnar1* gene thereby masking the latter phenotype. To test this hypothesis, we established EL4 tumors in *Ifnar1* KO mice. These mice displayed markedly accelerated tumor growth as compared to WT mice (Supplementary Fig. 3C) and had the same proportion and an absolute number of spleen PMN-MDSC and M-MDSC as WT TB mice (Fig. 2e). PMN-MDSC isolated from spleen and tumors of WT and *Ifnar1* KO EL4 TB mice had the same potent suppressive activity (Fig. 2f) suggesting that deletion of IFNAR1 is not sufficient to make MDSCs more suppressive in TB mice, since they may have already reached the maximum of immune suppression.

Then we asked whether the prevention of IFNAR1 downregulation could affect the suppressive activity of MDSC. To address this question, we used the knock-in mice that express the IFNAR1^S526A mutant protein, which is resistant to ubiquitination and degradation because it lacks critical Ser526, whose phosphorylation enables the recruitment of β-TrCP E3 ligase[51,52]. All tissues of these animals harbor homozygous *Ifnar1*^S526A alleles (*Ifnar1*^SA or SA). Naïve *Ifnar1*^SA mice did not exhibit an overt phenotype (this study and ref. [53]). MDSC from SA TB mice had substantially higher expression of IFNAR1 than cells from wild-type (WT) TB mice (Supplementary Fig. 3D). As expected, PMN-MDSC from SA TB mice had markedly higher expression of *Irf7* (Supplementary Fig. 3E).

Consistent with previous observations[33,54], tumors grew more slowly in SA mice than in WT mice (Fig. 3a). Hematopoietic cells were responsible for the observed delay in tumor growth since the reconstitution of lethally irradiated WT recipient mice with bone marrow from SA mice showed decreased tumor growth as compared to mice reconstituted with WT bone marrow cells (Fig. 3b). The proportion of PMN-MDSC in spleens of SA mice was reduced, however, the total number of PMN-MDSC and M-MDSC in SA and WT TB mice was comparable (Fig. 3c). In contrast to WT TB mice, neither spleen nor tumor PMN-MDSC in SA TB mice suppressed T-cell responses (Fig. 3d). Suppressive activity of M-MDSC was also completely abrogated in SA TB mice (Fig. 3e). To determine if the decrease in tumor progression in SA mice was mediated by CD8⁺ T cells, we treated mice with anti-CD8 antibodies and observed that depletion of these cells abrogated differences in tumor growth (Fig. 3f). However, depletion of CD8⁺ T cells did not affect the decrease in the proportion of PMN-MDSC observed in SA mice (Fig. 3g) and did not restore the suppressive activity of PMN-MDSC in SA mice (Fig. 3h). Thus, sustained IFNAR1 signaling due to preserved expression of the receptor in MDSC notably undermined suppressive activity. These results suggest that downregulation of IFNAR1 is required but not sufficient for pathologic activation of MDSC and for their ability to acquire immune-suppressive functions.

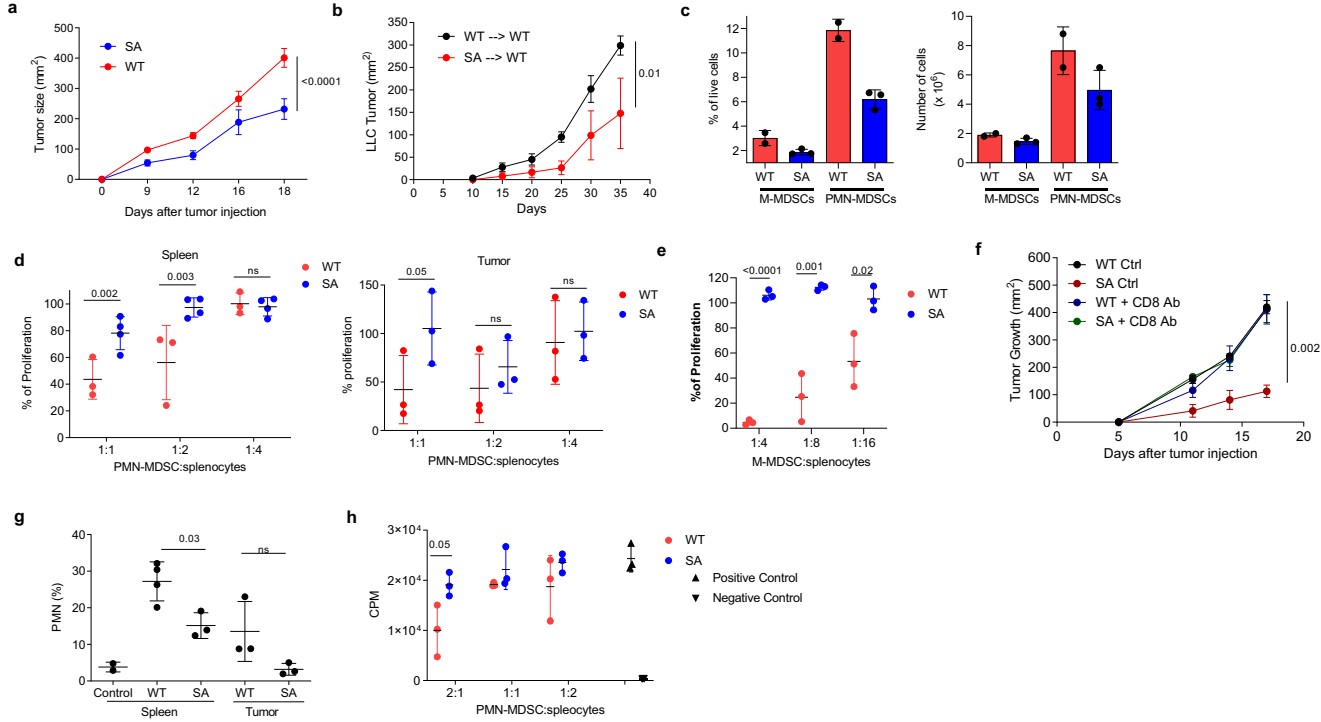

**Fig. 3 Stabilization of IFNAR1 abrogates MDSCs immune suppressive activity. a** EL-4 tumor growth in C57BL/6 WT and SA mice ($n = 4$). $p$ Value is calculated in a two-way Anova test. **b** LLC tumor growth in mice reconstituted with WT or SA BM cells ($n = 4$). $p$ Value is calculated in a two-way Anova test. **c** Percentage and an absolute number of M-MDSCs and PMN-MDSCs in the spleen of EL4 tumor-bearing mice ($n = 2$ for WT mice, $n = 3$ for SA mice). **d** Suppressive activity by PMN-MDSCs isolated from WT or SA spleen and tumors of EL4 tumor-bearing mice. Splenocytes from OT-1 mice stimulated with cognate peptide were used as effector cells. 100%—T cell proliferation without PMN-MDSCs; (spleen: $n = 3$ for WT mice, $n = 4$ for SA mice, tumor: $n = 3$ for all mice). $p$ Values were calculated in unpaired Student's $t$ test (two-sided for spleen and one-sided for tumor). **e** Suppressive activity by M-MDSCs isolated from the spleen of EL4 tumor-bearing mice. 100%—T cell proliferation without M-MDSCs; ($n = 3$). $p$ Values were calculated in unpaired two-sided Student's $t$ test. **f** EL4 tumor growth in WT and SA mice depleted of CD8+T cells. A representative of three independent experiments. $p$ Value is calculated in a two-way Anova test. $n = 3$ per group. **g** Percentage of PMN-MDSCs in spleen from WT and SA of EL4 tumor-bearing mice depleted of CD8+T cells ($n = 3$). $p$ Values were calculated in a one-way Anova test with correction for multiple comparisons. **h** Suppressive activity of HPC-derived PMN-MDSCs generated in vitro. T cell proliferation was evaluated in triplicate using [3H]-thymidine uptake (in counts per minute; CPM). Data are from one experiment representative of three experiments. $p$ Values were calculated using unpaired two-sided Student's $t$ test. In all panels, data are expressed as mean ± SEM are shown. ns not significant ($p > 0.05$).

**Mechanism of regulation of MDSC function by IFNAR1.** We next sought to delineate putative mechanisms by which downregulation of the IFNAR1 provides a license for immune-suppressive activities of MDSC. To this end, WT and SA EL4 TB mice were treated with CD8 antibodies to eliminate differences in tumor growth, PMN-MDSC were collected 18 days after tumor inoculation and expression of genes implicated in MDSC suppressive activity was evaluated. PMN-MDSC from SA TB mice had markedly lower expression of *Arg1* and downregulation of *Ptgs2*, and *Nox2* (Fig. 4a). ROS production is one of the hallmarks of PMN-MDSC activity. NOX2 is directly involved in the production of ROS by PMN-MDSC. Because of a substantial decrease in *Nox2* expression, we assessed the level of ROS in PMN-MDSC. Both spleen and tumor PMN-MDSC from SA TB mice demonstrated a markedly lower level of ROS than WT mice (Fig. 4b). These results suggest that downregulation of IFNAR1 on MDSC is required for their ability to express several critical mediators of the immune-suppressive activity.

To gain further insight into the mechanisms delineating the effects of IFN1 on suppressive activities, we performed RNA sequencing of PMN-MDSC from WT and SA EL4 TB mice. The most prominent differences in the expression of specific transcripts are shown in Fig. 4c. Pathway analysis revealed that stabilization of IFNAR1 in PMN-MDSC in TB mice resulted in significant inhibition of glycolysis in these cells, as well as several

pathways associated with cell movement and chemokine signaling (Fig. 4d). Thus, unabated IFNAR1 signaling was associated with the marked reduction of several major mediators of PMN-MDSC suppressive activity.

**Mechanisms regulating IFNAR1 expression in MDSC.** We next sought to identify the mechanisms regulating IFNAR1 expression on MDSC. It is known that the IFNAR1 level is regulated by phosphorylation-dependent ubiquitination and degradation of the IFNAR1 chain[39]. This proteolytic inactivation of IFNAR1 is mediated by either protein kinase D2 in response to IFN1 itself[55] or by p38 protein kinase in response to the non-ligand stimuli that are present in the tumor microenvironment including pro-inflammatory cytokines, tumor-derived vesicles, and the deficit of oxygen/nutrients leading to the integrated stress response[33,38,39]. Therefore, we tested the effect of tumor-derived factors on IFNAR1 expression by exposing mouse PMN or Mon to the tumor explant supernatant (TES) obtained from different EL-4 tumors. We observed marked downregulation of IFNAR1 in PMN and Mon treated with TES (Fig. 5a), which contained negligible (if any) amounts of IFNα and IFNβ (Fig. 5b) suggesting the role of non-ligand factors in decrease in IFNAR1 levels. Similar experiments were performed with PMN isolated from healthy donors. TES from different primary human tumors, as

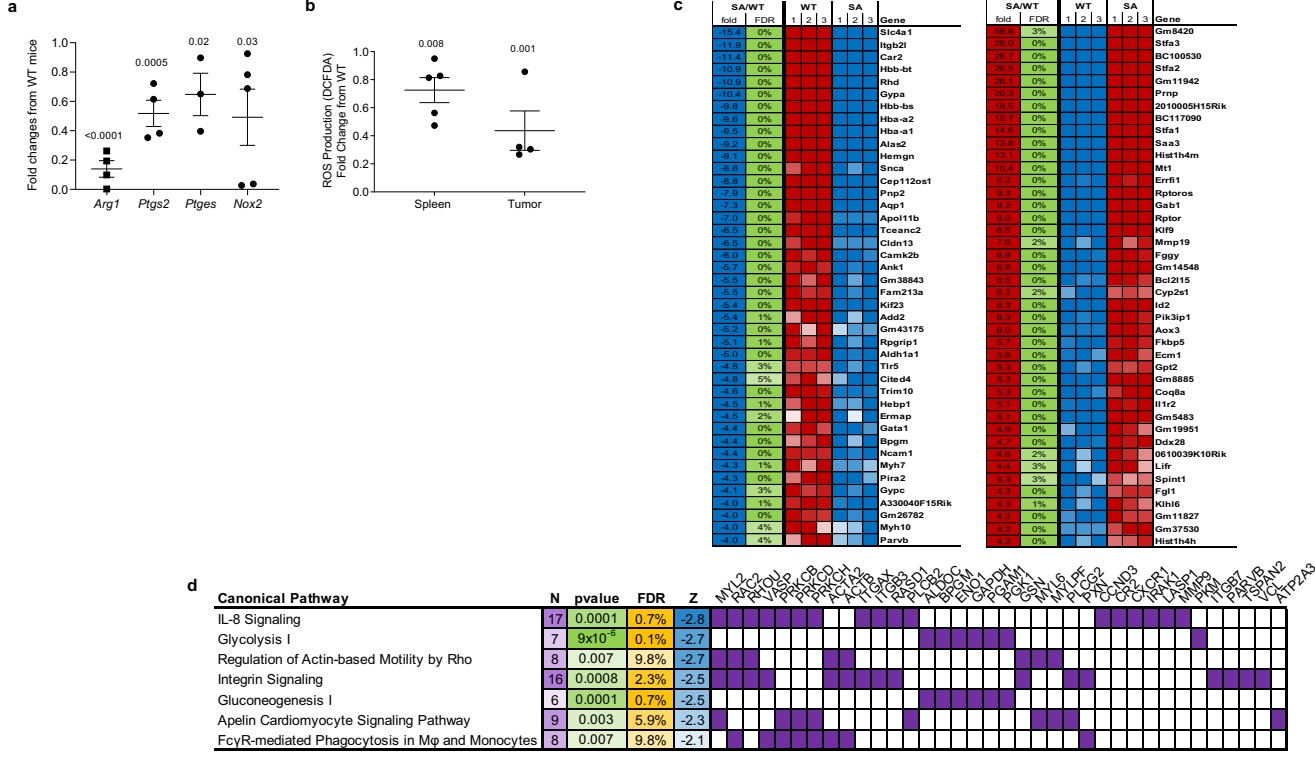

**Fig. 4 Effect of Ifnar1$^{SA}$ on mRNA expression gene expression by PMN-MDSCs. a** mRNA expression of genes associated with immunosuppressive activity in PMN-MDSCs. Analyzed by RT-qPCR. **b** Reactive oxygen species (ROS) levels by PMN-MDSCs from spleen and tumors of WT and *Ifnar1*$^{SA}$ tumor-bearing mice ($n = 4$). Data are expressed as mean ± SEM and the *p* values were calculated using unpaired two-sided Student's *t* test. **c** Expression heatmap of significantly affected genes (FDR < 5%) for different fold change thresholds. **d** List of canonical pathways downregulated that were determined by Ingenuity Pathway Analysis (IPA) as significantly enriched pathways affected by *Ifnar1*$^{SA}$. N = number of genes; *p* value = enrichment; FDR = false discovery rate of significant genes (FDR < 10%) for different fold change thresholds; Z = activation z-scores calculated by IPA represent predicted canonical pathway decreasing (Z threshold of 2 is used to call the state). A panel of top genes affected by *Ifnar1*$^{SA}$ for each pathway. Fisher exact test was used to calculate FDR.

well as supernatant from tumor cell line PCI-30, caused substantial decreases in IFNAR1 expression (Fig. 5c). TES also caused downregulation of IFNAR1 in PMN, but not Mon generated from CD34$^+$ progenitor cells (Fig. 5d).

Thus, tumor-derived factors present in TES from both mouse and human tumor cells caused downregulation of the receptor. This may explain the decrease of IFNAR1 in PMN-MDSC from peripheral lymphoid organs. However, since the downregulation of IFNAR1 was markedly stronger in tumors than in spleens, we asked whether anything could further downregulate IFNAR1 in MDSC in the tumor site. Hypoxic conditions downregulated IFNAR1 in melanoma cells[37] and hypoxic areas of solid tumors lacked IFNAR1 expression[33]. Exposure of mouse PMN to hypoxia or lactic acid (major contributors to endoplasmic reticulum (ER) stress response) notably decreased the receptor levels (Fig. 5e). We and others previously implicated ER stress in the immune-suppressive activity of MDSC[56,57]. Therefore, we explored the role of ER stress in the regulation of IFNAR1 in MDSC. To model the intratumoral stress, we used a potent ER stress inducer thapsigargin (THG), which caused marked a reduction in IFNAR1 expression in mouse PMN and Mon (Fig. 5f). Previously, we showed that THG induces suppressive activity in PMN. Therefore, we tested the possibility that THG-induced suppression can be mediated via IFNAR1. PMN were isolated from WT and SA mice and then treated with THG. THG caused potent but similar levels of suppressive activity in PMN from WT and SA mice (Fig. 5h). In WT mice pre-treatment of PMN with INFβ did not prevent induction of suppressive activity

by THG (Fig. 5i). Thus, IFNAR1 expression was not sufficient to control potent ER stress induction by THG.

Although IFN signaling can be mediated by IFNAR2, the current paradigm indicates that IFN signaling is impossible without IFNAR1[58–64]. Nevertheless, we tested the possible role of IFNAR2 in the conversion of PMN to PMN-MDSC. First, we quantified IFNAR2 expression on human PMN after the treatment with TES. In contrast to IFNAR1, which was down-regulated by TES (Fig. 5c) IFNAR2 expression was not affected (Supplementary Fig. 4A). Furthermore, we obtained bone marrow cells from *Ifnar2* deficient mice[65] and performed the experiments to determine the possibility that lack of IFNAR2 alone could drive the conversion of PMN in PMN-MDSC. No suppression was detected (Supplementary Fig. 4B).

We focused on the mechanisms underlying IFNAR1 down-regulation in MDSC. Since p38 was implicated in the ligand-independent IFNAR1 ubiquitination and degradation[35,37,38], we investigated the role of this kinase. Deletion of p38 completely prevented downregulation of IFNAR1 in spleen PMN-MDSC (Fig. 6a) suggesting an important role of p38 kinase in regulation of IFNAR1 on MDSC.

Treatment of mouse PMN with TES activated p38 (as evident by its increased phosphorylation) (Fig. 6b). Tumor PMN-MDSC had higher levels of phospho-p38 than spleen PMN-MDSC (Fig. 6c). As expected, hypoxia increased phospho-p38 in PMN (Fig. 6d). In the absence of p38 (in p38 KO mice), expression of IFNAR1 was dramatically higher in PMN treated with TES alone or in combination with hypoxia (Fig. 6e). Since PMN-MDSC

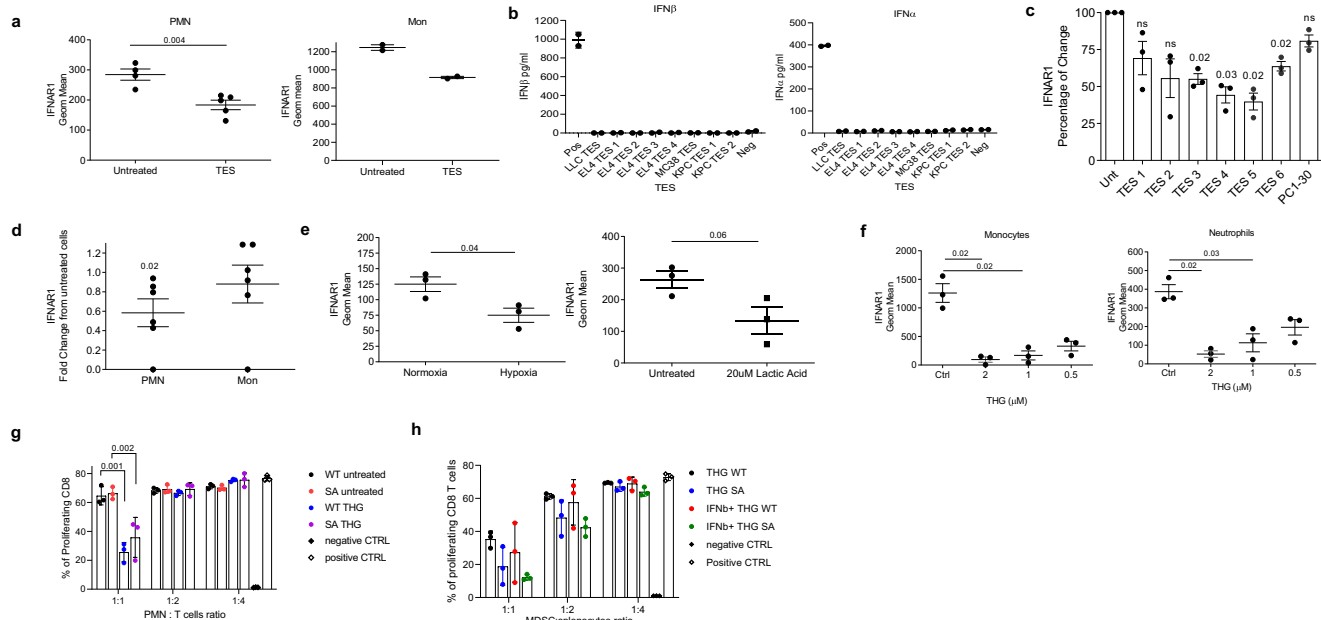

**Fig. 5 Tumor microenvironment induces downregulation of IFNAR1 in MDSCs. a** IFNAR1 levels on bone marrow PMN (left) or Mon (right) treated for 18 h with 20% TES in vitro. ($n = 4$ untreated PMN, $n = 5$ TES treated PMN, $n = 2$—Mon). p Value was calculated in unpaired two-sided Student's t test. **b** IFNβ and IFNα amount in eight different types of TES measured by ELISA. **c** Expression of IFNAR1 on PMNs, isolated from healthy donors' blood ($n = 3$), treated for 16–18 h with 6 different TES (30%) and 30% PCI-30 tumor-conditioned media (TCM) in vitro. $N = 3$. p Values were calculated in a one-way Anova test with correction for multiple comparisons. **d** IFNAR1 levels on CD34-derived PMNs or Mon treated with 30% TES at day 7 followed by flow cytometry analysis on day 8 ($n = 6$). **e** BM PMNs ($n = 3$) were cultured in 0.5% $O_2$ (hypoxia) or 20 μM lactic acid for 16–18 h before measuring IFNAR1 levels by flow cytometry. In **d**, **e**, p values were calculated in unpaired two-sided Student's t test. **f** IFNAR1 levels on BM PMNs or Mon treated for 16–18 h with different doses of ER stress inducer thapsigargin (THG; 2 μM, 1 μM, or 0.5 μM) in vitro ($n = 3$). p Values were calculated in a one-way ANOVA test with corrections for multiple comparisons. **g** PMN were isolated from BM of naïve WT or SA mice, treated overnight with 1 μM of THG. THG has then washed away and suppressive activity of the cells was measured by coculturing PMN with PMEL splenocytes stained with cell trace Far-red at different ratios and stimulated with the cognate peptide (gp100, 10 ng/ml). Cell trace dilution was measured after 48 h to assess the proliferation of CD8+ T cells ($n = 3$). p Values were calculated in one-way ANOVA test with corrections for multiple comparisons. **h** PMN were isolated from BM of naïve WT or SA mice pretreated for 2 h with 2000 U/ml IFNβ then overnight with 1 μM of THG. Suppressive activity of the cells was measured by coculturing PMN with PMEL splenocytes stained with Cell trace Far-red and stimulated with the cognate peptide (gp100, 10 ng/ml). Cell trace dilution was measured after 48 h to assess the proliferation of CD8+ T cells ($n = 3$). In all figures, data are expressed as mean ± SD.

demonstrated higher downregulation of IFNAR1 than M-MDSC we asked if p38 was activated similarly in these cells. PMN and monocytes were isolated from naïve tumor-free mice and treated with TES. Monocytes had a higher basal level of p-p38 than neutrophils and it is only slightly increased by TES treatment as compared to neutrophils where TES caused substantial upregulation of p-p38 (Fig. 6f). These results implicate p38 kinase in the regulation of myeloid cell responses to the tumor-derived factors.

We tested the effect of THG and tumor cell-conditioned medium on ubiquitination and degradation of IFNAR1 in THP-1 myeloid cell line. Both, THG and tumor cell-conditioned medium triggered phosphorylation and ubiquitination of IFNAR1 while decreasing total levels of IFNAR1 protein. Inhibitor of p38 kinase LY2228820 (Ralimetinib) abrogated those effects (Fig. 6g) confirming the role of p38 in ubiquitination and degradation of IFNAR1 caused by tumor-derived factors and ER stress in myeloid cells.

Treatment of human PMN with TES that caused downregulation of IFNAR1 (Fig. 5c) induced upregulation of phospho-p38 (Fig. 6h). These data collectively suggest that activation of p38 kinase by TES or hypoxia mediates downregulation of IFNAR1 on MDSC. It is important to point out that phosphorylation of p38 is regulated by many nonredundant factors. Hypoxia and soluble factors present in TES are only several of many possible mechanisms present in the tumor microenvironment.

**Therapeutic regulation of p38 abrogated suppressive activity of MDSC and elicits the antitumor effects**. These results suggest that inhibition of p38 kinase may prevent downregulation of IFNAR1 expression and ensuing acquisition of suppressive activities by MDSC. To test this possibility, we used the p38 kinase inhibitor LY2228820. In healthy donor PMN, treatment with p38 kinase inhibitor upregulated expression of IFNAR1 and downregulation of IFNAR1 caused by TES was abrogated by inhibition of p38 kinase (Fig. 7a). Treatment with p38 kinase inhibitor caused upregulation of IFNAR1 in mouse PMN but not Mon (Fig. 7b). TES-induced downregulation of IFNAR1 expression was abrogated by inhibition of p38 kinase (Fig. 7b). Administration of LY2228820 inhibited the suppressive activity of PMN from control mice treated with TES (Fig. 7c) as well as spleen PMN-MDSC from EL4 TB mice (Fig. 7d). In contrast, treatment of control PMN with LY2228820 alone did not affect their ability to regulate T cell proliferation (Fig. 7e). These combined results suggest that a p38 kinase inhibitor may show an enhanced antitumor effect when combined with an IFN1 inducer. To test this hypothesis, TB mice were treated with Poly:IC (inducer of IFN1) and LY2228820. Treatment with Poly:IC alone or p38 inhibitor alone did not have a substantial effect on the proportion or an absolute number of myeloid cells, PMN-MDSC, M-MDSC, macrophages, or DCs in the spleen. Combination therapy caused only a modest increase in the number of DCs but had minimal effect on other myeloid cell populations

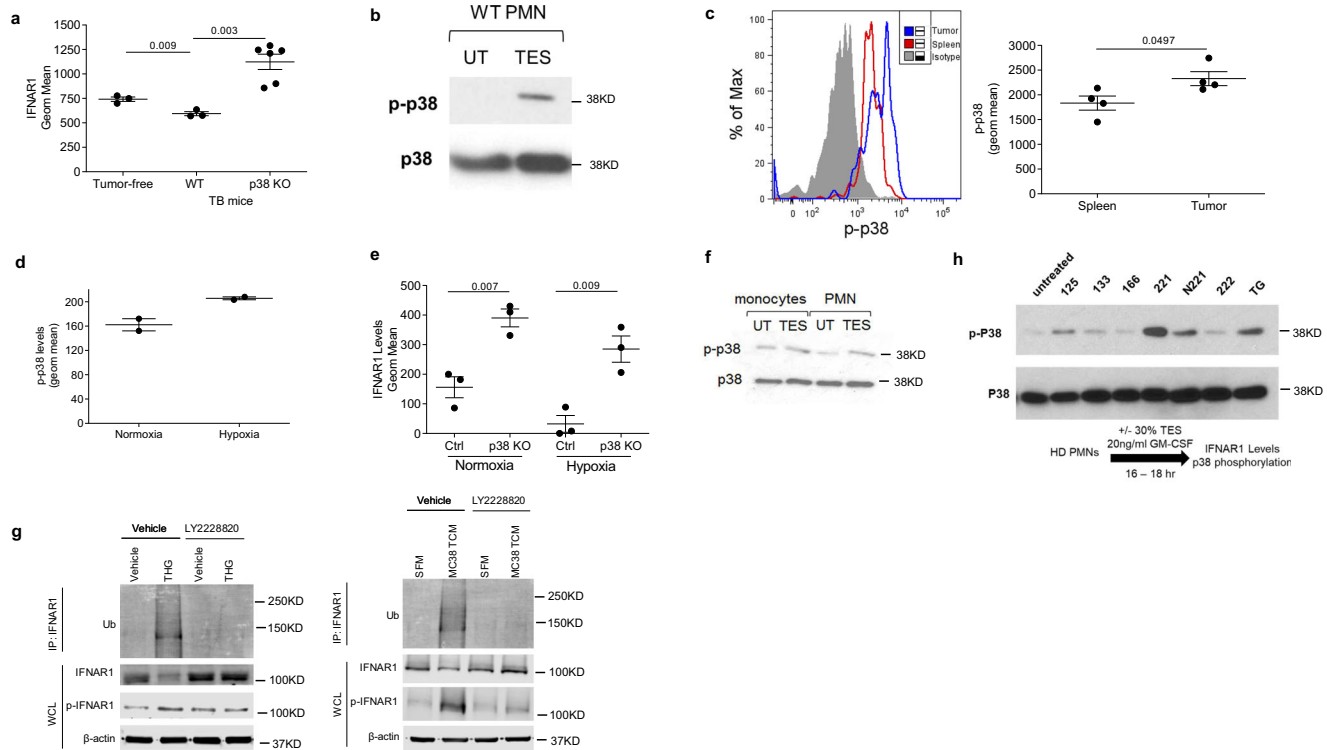

**Fig. 6 Downregulation of IFNAR1 in MDSCs is mediated by p38. a** IFNAR1 levels on PMN-MDSCs from tumor-free or WT and p38 KO (*Mapk14*$^{\Delta/\Delta}$) TB mice ($n = 3$ for tumor-free and WT groups, $n = 6$ for p38 KO group). Data are expressed as mean ± SEM and the *p* values were calculated using unpaired two-sided Student's *t* test. **b** Phosphorylation and total p38 protein in BM PMNs treated with 20% TES for 18 h in vitro. A typical example of two performed experiments is shown. **c** Histogram (left) of phosphorylation of p38 (p-p38) in spleen (red) and tumor (blue) PMN-MDSCs compared to isotype (gray) ($n = 4$; right). *p* Value was calculated in unpaired two-sided Student's *t* test. **d** BM PMNs were cultured in 0.5% $O_2$ for 2 h and p-p38 was measured by flow cytometry ($n = 2$). **e** IFNAR1 levels on BM PMNs from WT or p38 KO cultured with 30% TES in normoxia or hypoxia for 16–18 h ($n = 3$). Data are expressed as mean ± SEM and the *p* values were calculated using unpaired two-sided Student's *t* test. **f** Monocytes and PMN were isolated from bone marrows from naïve mice, treated for 18 h in vitro with 20% of TES. Cells were then lysed and phospho-p38 and total p38 measured by western blot. Experiments were performed twice with the same result. **g** Totally, $1 \times 10^7$ of THP-1 cells were pretreated with vehicle (DMSO) or p38 inhibitor LY2228820 1 h prior to the 1 h treatment with 1 μM THG or 30 min treatment with MC38 tumor conditioned media (75%, v/v). Serum-free medium (SFM) served as control. Cell lysates were immunoprecipitated with IFNAR1 antibody and probed with anti-ubiquitin antibody. Whole-cell lysates (WCL) were used to evaluate IFNAR1 and p-IFNAR1 proteins. Two experiments with the same results were performed. **h** Phosphorylated and total p38 protein in HD PMNs treated with different TES (30%) or THG (1 μM) for 16–18 h in vitro.

(Supplementary Fig. S5). Treatment of TB mice with Poly:IC and LY2228820 alone had very minor effect on tumor growth. However, a combination of these compounds elicited a substantial antitumor activity (Fig. 7f). This effect was abrogated by treatment with an anti-CD8 antibody (Fig. 7g). When the suppressive activity of PMN-MDSC was compared in these mice (with the same tumor burden), only PMN-MDSC from untreated TB mice were able to inhibit T-cell proliferation, whereas no suppressive activity was observed in PMN-MDSC isolated from treated mice (Fig. 7h). To test the association of the therapeutic effect of p38 inhibitor and poly:IC with IFNAR1 expression, we performed treatment experiments in mice with targeted deletion of IFNAR1 in neutrophils and monocytes (IFNAR1$^{fl/fl}$S100A8$^{Cre}$). We observed no antitumor effect of poly-I:C and LY2228820 treatment in those mice (Fig. 7i). These results clearly indicate that the effect of p38 inhibitor was indeed mediated in large part by IFNAR1 expression on MDSC. Thus, stabilization of IFNAR1 by inhibition of p38 kinase resulted in antitumor effect in combination with type I interferon induction.

## Discussion

In this study, we identified the role of IFN1 in restricting the acquisition of immune-suppressive activity by MDSC.

Furthermore, our data indicated that tumor-derived factors-driven p38 kinase-dependent downregulation of IFNAR1 represents a mechanism, by which this role of the IFN1 pathway in myeloid cells is overwhelmed during tumorigenesis leading to the acquiescence of immune-suppressive activities.

Only a proportion of PMN in cancer patients acquires features of MDSC[25]. The presence of PMN-MDSC and M-MDSC in cancer patients is correlated with a negative clinical outcome[6]. It indicates, that at any given moment, populations of classical PMN and Mon co-exist with immune suppressive PMN-MDSC and M-MDSC. What would prevent all PMN and Mon from acquiring characteristics of MDSC? We hypothesized that there could be a mechanism that restricts the acquisition of immune-suppressive activity by PMN and Mon. Our data suggest that whereas the IFNAR1-mediated IFN1 signaling may be such a mechanism, this pathway is inactivated during tumorigenesis.

Our recent work demonstrated that IFNAR1 was downregulated on all types of cells in the tumor microenvironment of colorectal cancers leading to inactivation of the IFN1 pathway and generation of immune-privileged niches[32,33]. We found that downregulation of IFNAR1 on MDSC was not restricted to the tumor site, it was observed in PB of cancer patients and in spleens of TB mice. However, IFNAR1 downregulation was higher in tumor MDSC than in spleen MDSC. It was associated with the more

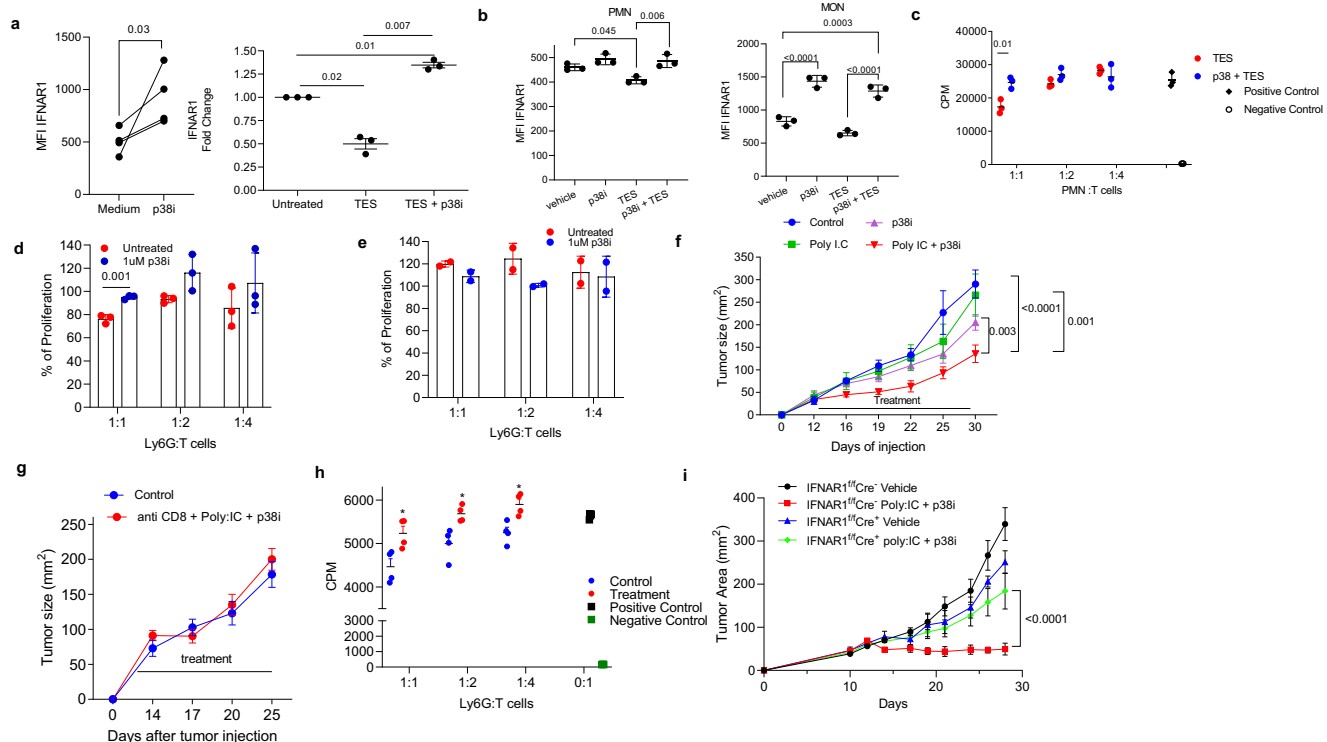

**Fig. 7 Effect of pharmacological inhibition of p38 phosphorylation of MDSC function and tumor growth. a** Left panel. PMN were isolated from the blood of healthy volunteers and cultured for 16–18 h with medium containing 20 ng/ml of GM-CSF with or without p38i (LY2228829 1 µM). IFNAR1 level was measured by flow cytometry. N = 4. p Value was calculated in two-sided unpaired Student's t test. Right panel. Healthy donors PMNs were pre-treated with 1 µM LY2228820 (p38 inhibitor; p38i) followed by 16–18 h incubation with 30% TES. IFNAR1 levels were measured by flow cytometry (n = 3) and expressed as fold changes over untreated cells. p Values were calculated in one-way ANOVA test with correction for multiple comparisons. **b** PMN and Monocytes were isolated from the bone marrow of mice by cell sorting, pretreated with 1 µM p38i followed by incubation with 20% TES (n = 3) for 16–18 h. IFNAR1 was measured by flow cytometry. Geometric MFI is shown. Data are expressed as mean ± SEM. p Values were calculated in one-way ANOVA test with corrections for multiple comparisons. **c** HPC-derived PMN-MDSCs were treated with 1 µM p38i before assessing their suppressive activity in triplicates. Two experiments with the same results were performed. Data are expressed as mean ± SEM (n = 3). p Values were calculated in two-sided unpaired Student's t test. **d** Suppressive function of PMN-MDSCs from the spleen of EL-4 TB mice after in vitro treatment for 3 h with 1 µM p38i before coculture with responder T cells. p Value was calculated in two-sided unpaired Student's t test. **e** BM PMNs from tumor-free mice were treated with 1 µM p38i for 3 h prior to the suppression assay. **f** MC38 tumor growth in mice treated with Poly I:C and p38i (n = 4). p Values are calculated by two-way ANOVA test with correction for repeated measurements. **g** MC38 tumor growth in mice treated with anti-CD8 antibody, Poly I:C and p38i (n = 5). Control is mice treated with PBS. p Values are calculated by two-way ANOVA test with correction for repeated measurements. **h** Suppressive activity of spleen PMN-MDSCs from control or anti-CD8 antibody, Poly I:C and p38i treatment group (n = 4). p Values were calculated in unpaired, two-sided Student's t test. *p < 0.05. **i** Totally, $10^6$ MC38 cells was injected s.c. to the mice lacking IFNAR1 in myeloid cells (Ifnar1$^{fl/fl}$ Cre$^+$) or control mice (Ifnar1$^{fl/fl}$ Cre$^-$), and tumor growth was measured every 2–4 days (n = 5 in Ifnar1$^{fl/fl}$ Cre$^-$ group, n = 4 in all other groups). Treatment started at day 12 with poly: IC in PBS (10 µg/mouse) i.p. daily and p38 inhibitor (LY2228829 1 mg/kg) prepared in methylcellulose administered by oral gavage every other day. Control mice received an equal amount of vehicle (PBS i.p. and methylcellulose by oral gavage). p Values are calculated by two-way ANOVA test with correction for repeated measurements.

potent suppressive activity of MDSC in tumors than in spleens. This was consistent with previous observations[66–69] and suggested that downregulation of IFNAR1 and suppressive activity of MDSC could be linked. The important role of IFN1 in MDSC suppressive activity was established directly in two sets of experiments: suppressive activity of MDSC generated by exposure to tumor-derived factors present in TES and caused by ER stress inducer thapsigargin. In both experimental systems, IFNβ blocked the suppressive abilities of MDSCs. Whereas in tumor-free mice, deletion of IFNAR1 did not suffice to convert PMN or Mon to suppressive MDSC, the latter is facilitated by the presence of additional positive signals, for instance, activation of STAT3, NF-kB, or other mechanism mediated by tumor-derived factors, as described in the previous studies[70]. However, expression of IFNAR1 was required to prevent this conversion and led to the inactivation of tumor-suppressive activities. This conclusion is based on the experiments with Ifnar1$^{S526A}$ (SA) mice that had a stable expression of the

receptor. MDSC in these mice lost suppressive activity. These data suggest that downregulation of the receptor is one of the major mechanisms that inactivate the IFN1 pathway in MDSC and enable their immune-suppressive properties.

Our data also suggest a mechanism by which IFN1 regulates the activities of MDSC. Presented here data show that downregulation of IFNAR1 contributes to the expression of Arg1 and Nox2, two critical mediators of MDSC suppressive activities. Given that these mediators are expressed downstream of STAT3[71,72], whose transcriptional activity is suppressed by IFN1[73], it is plausible that downregulation of IFNAR1 contributes to maintaining STAT3 activities and relevant STAT3-driven immune suppression. Additional mechanisms hinted at by results of transcriptional profiling also cannot be ruled out and merit further investigation.

What could cause IFNAR1 downregulation in MDSC? Our data demonstrated that TES causes decreased IFNAR1 expression

in human and mouse PMN and Mon. Since neither IFNα nor IFNβ was detectable in those TES, ligand-independent downregulation of IFNAR1 was more likely. TES contains multiple cytokines and tumor-derived extracellular vesicles that can cause pathological MDSC activation[74–76]. In addition, cells from SA mice were shown to be relatively deficient in the uptake of the tumor-derived extracellular vesicles[36], which may partially explain the lack of suppressive activity. However, it is unlikely that one or even a few of these mostly redundant factors are solely responsible for this phenomenon. We found that deletion of p38 completely abrogated downregulation of IFNAR1 in MDSC. Similar results were obtained with selective p38 inhibitor, strongly suggesting that p38 activation in response to diverse tumor-derived factors may be a major mechanism that regulates IFNAR1 expression in MDSC.

Our results suggest that the expression of IFNAR1 on MDSC may be a critical mechanism regulating their suppressive activity. Pharmacological regulation of the receptor by inhibiting p38 activation substantially enhances the antitumor effect of IFN1 inducers and thus may open therapeutic opportunities.

## Methods

**Human samples.** PB was collected from untreated cancer patients at the Helen F. Graham Cancer Center. The study was approved by the institutional review boards (IRBs) of the Christiana Care Health System at Helen F. Graham Cancer Center and The Wistar Institute. All the patients and healthy donors signed IRB-approved consent forms. PB was collected from (i) 12 patients with non-small cell lung cancer and 5 with small cell lung cancer; (ii) 5 patients with breast cancer, (iii) 10 patients with colorectal cancer, (iv) 4 patients with pancreatic adenocarcinomas, (v) 3 patients with esophageal cancer; (vi) 3 patients with head and neck cancer, (vii) 1 patient with gastric cancer, and (viii) 1 patient had renal cancer. In some patients with lung, colorectal, renal, or head and neck cancer, tumor tissues (0.2–1 g) were surgically removed. The ages of cancer patients were between 46 and 92 years (median, 70 years), 24 males and 20 females. Peripheral sample of blood from 21 healthy donors with ages 30–64 (median, 55 years), 7 males 14 females and were used as a control for cancer patients or for in vitro experiments.

**Mice.** All experiments with animals were approved by the IACUC of The Wistar Institute. Balb/c or C57BL/6 mice (female, 6–8 weeks old) were obtained from Charles River. OT-I TCR-transgenic mice (C57Bl/6-Tg(TCRaTCRb)1100mjb) (female, 6–8 week old) and Pmel mice (B6.Cg-Thy1$^a$/Cy Tg(TcraTcrb)8Rest/J) were purchased from Jackson Laboratory. Littermate C57BL/6 ("WT"), C57BL/6 Ifnar1$^{S526A}$ mice ("SA"), Ifnar1$^{-/-}$ and Ubc9-CreER::Mapk14$^{f/f}$ mice were described previously[33]. Ifnar1$^{f/f}$S100A8$^{Cre}$ mice were generated by crossing Ifnar1$^{fl/}$$^{fl}$ mice with S100A8$^{Cre}$ miceRET melanoma were obtained from Dr. Umansky (German Cancer Center, Heidelberg, Germany). KPC mice were obtained from Dr. Robert Vonderheide (University of Pennsylvania). Ifnar2$^{-/-}$ mice (Ifnar2tm1 (KOMP)Vlcg), which were originally purchased from UC Davis KOMP Repository were bred and maintained at Montana State University (Bozeman, MT) according to the IACUC regulations. Bone marrow cells from these mice were provided by Dr. Rynda-Apple.

**Bone marrow chimeras.** Mixed bone marrow (BM) chimeric mice were obtained as described previously[77]. Briefly, pooled tibial and femoral BM cells from donor mice were lysed with ACK buffer. BM cells from SA mice transferred to lethally irradiated syngeneic (CD45.1$^+$) recipients obtained from Charles River. In control BM from WT mice was transferred to lethally irradiated recipients. Totally, 8–10 weeks after BM reconstitution mice were injected s.c. with $5 \times 10^5$ LLC cells and measured tumor growth.

**Cell lines and tumor models.** *Mouse cell lines*: EL4 lymphoma, LLC (Lewis lung carcinoma), CT-26 colon carcinoma, B16F10 melanoma were purchase from ATCC. MC38 colon carcinoma (provided by I. Turkova, University of Pittsburgh, Pittsburgh, PA). MethA (methylcholantrene induced) sarcoma cell line was originally obtained from Dr. Lloyd J. Old (Cancer Research Institute, New York, NY). MethA tumor was established i.p. in Balb/c mice and passaged in vivo as an ascitic tumor. LLC or B16F10 and CT-26 TB mice were generated by injecting $5 \times 10^5$ tumor cells s.c. into C57BL/6 or Balb/c mice, respectively. EL-4 and MethA TB mice were generated by injecting $5 \times 10^5$ tumor cells i.p. into C57BL/6 mice. We harvested cells from ascites tumors by washing the peritoneum with 10 ml of ice-cold MACS buffer. For MC38 TB mice, we injected $1 \times 10^6$ tumor cells s.c. into C57BL/6 mice.

*Human cell lines*. PCI-30 (head and neck squamous carcinoma) was obtained from ATCC.

**Isolation of human cells.** Human PMNs from healthy donors were isolated with two methods. (1) PMNs were isolated by centrifugation using double density gradient Histopaque (Sigma). PBMCs were collected using 1.077 and PMNs were collected using 1.119. (2) Whole blood was enriched for PMNs using MACSxpress Neutrophil Isolation Kit (Miltenyi) following the protocol provided by the manufacturer.

**Flow cytometry.** *Mouse.* Single-cell suspensions of BM, spleen, and tumors were prepared, and red cells were removed using ACK lysing buffer. In other experiments, cells were culture in vitro before flow cytometry analysis was performed. All antibody incubations were performed for 15 min at 4 °C in dark and all centrifugation was done at 1500 r.p.m. at 4 °C for 5 min. Usually, up to $1 \times 10^6$ cells were incubated with Fc-block (BD Biosciences) for 10 min and surface staining was performed at 4 °C for 15 min. Cells were run on an LSRII flow cytometer (BD Biosciences) and data were analyzed by FlowJo (Tristar).

*Human.* Single-cell suspensions from peripheral blood were incubated with Fc-block (Miltenyi) for 10 min and surface staining was performed at 4 °C for 30 min. Cells analyzed as described above. A list of antibodies used is in Supplementary Table 1.

**Isolation of PMN-MDSCs and M-MDSCs.** Single-cell suspensions were prepared from the spleen followed by red blood cell removal using ACK buffer. Single-cell suspensions from tumor tissues were prepared using Mouse Tumor Dissociation Kit according to the manufacturer's recommendation (Miltenyi). For PMN-MDSCs isolation, cells were labeled with biotinylated anti-Ly6G antibody (Miltenyi Biotec), incubated with streptavidin-coated microbeads (Miltenyi Biotec), and separated on MACS columns (Miltenyi Biotec). M-MDSCs were sorted by using either FACS Aria II (BD Biosciences) and MoFlo Astrios EQ (Beckman Coulter) cell sorters. The antibodies for M-MDSCs isolation are described in Supplementary Table 1.

**Suppression assay.** *Mouse.* After isolation of Ly6G$^+$ or Ly6C$^+$ cells as described above, cells were plated in U-bottom 96-well plates (duplicates or triplicates) in complete RPMI. Cells were co-cultured at various ratios with total splenocytes from Pmel or OT-1 transgenic mice in the presence of cognate peptides: OT-1 (SIINFEKL; 0.05 ng/ml) and Pmel (EGSRNQDWL; 0.1 μg/ml). After 48 h, cells were incubated with [$^3$H]-thymidine (PerkinElmer) for 16–18 h. Proliferation was measured by using TopCount NXT instrument (PerkinElmer). *Human.* In vitro treated PMNs from healthy donors were plated in U-bottom 96-well plates (triplicates) in complete RPMI. Concurrently, CD3$^+$ T cells were isolated from PBMCs of the same donor using the EasySep Human T Cell Enrichment Kit (STEMCELL Technologies). PMNs were coculture at different ratios with $10^5$ T cells and 2.5 μl of Human T-Activator CD3/CD28 Dynabeads (Gibco). After 48 h, cells were incubated with [$^3$H]-thymidine as described above.

**Generation of TES.** *Mouse.* TES were prepared from excised non-ulcerated EL4, LLC, MC38, Ret melanoma, or KPC pancreas tumors. A small tumor piece (0.5 g) was harvested, minced into pieces <3 mm in diameter and resuspended in complete RPMI. After 16–18 h of incubation at 37 °C with 5% CO$_2$, the cell-free supernatant was collected using 0.22 μm filters (EMD Millipore) and kept at −80 °C.

*Human.* TES was prepared from surgically removed tumors and a small tumor piece (0.1–0.5 g) was processed. After 16–18 h of incubation at 37 °C with 5% CO$_2$, the cell-free supernatant was collected using 0.22 μm filters (EMD Millipore) and kept at −80 °C.

**ELISA.** Mouse interferon-alpha (IFN-α) and beta (IFN-β) concentrations in TES were measured by using Mouse IFN Alpha ELISA Kit (TCM) (PBL Assay Science) and Mouse IFN-beta ELISA Kit (R&D Systems), according to manufacturer's instructions.

**Western blot analysis.** Proteins were extracted from BM or HD PMNs in RIPA buffer followed by western blot staining with anti-p38 (Santa Cruz Biotech) and anti-phospho-p38 (Cell Signaling) followed by anti-rabbit-HRP conjugated secondary antibodies (Sigma Aldrich).

**MDSCs generated in vitro.** *Mouse.* Hematopoietic progenitor cells (HPCs) were isolated from mouse bone marrow by using a Lineage depletion kit (Miltenyi), according to the manufacturer's instructions. Cells were seeded at 25,000 cell/ml in 24-well plates and recombinant GM-CSF (20 ng/ml; Invitrogen), 20% v/v TES were added on day 1 and day 3. At day 5, Ly6G positive neutrophils were isolated by using anti-Ly6G biotin (Miltenyi) and streptavidin beads (Miltenyi), according to manufacturer's followed by suppression assay. In addition, total cells were stained and analyzed for flow cytometry. In other experiments, LY2228820 p38 inhibitor (1 μM; Selleckchem) was added to HPC culture on day 3.

*Human.* Hematopoietic progenitor cells (HPCs) were isolated from cord blood using the CD34 MicroBead Kit (Miltenyi), according to the manufacturer's instructions. Cells were seeded at $5 \times 10^4$ cell/ml in 6-well plates with recombinant

G-CSF (100 ng/ml; PeproTech) and GM-CSF (10 ng/ml; PeproTech). At day 7, 30% v/v TES were added and the next day flow cytometry analysis was performed.

**In vitro treatment.** *Mouse.* Splenic PMN-MDSCs or M-MDSCs isolated from TB mice were treated with mouse IFNβ (2000 units/ml; PBL Assay Science) for 2 h before assessing their suppressive activity. Total BM cells were treated with TES (30% v/v) or thapsigargin (THG) (0.5, 1, 2 μM; Sigma) and recombinant GM-CSF (10 ng/ml) for 16–18 h followed by measurement of cell surface IFNAR1 levels by flow cytometry. BM PMNs treated with lactic acid (20 μM; Sigma Aldrich) and recombinant GM-CSF (10 ng/ml) for 16–18 h followed by flow cytometry analysis. BM PMNs and Mon pretreated with LY2228820 p38 inhibitor (1 μM; Selleckchem) or vehicle (DMSO) for 2 h and then treated with TES (30% v/v) for an additional 2 h. Experiments with hypoxia (0.5% $O_2$) for 16–18 were maintained using a hypoxic chamber (BioSpherix).

*Human.* HD PMNs were isolated as described above and treated with TES (30% v/v), TCM (30% v/v) or THG (1 μM; Sigma), and recombinant GM-CSF (20 ng/ml; PeproTech) for 16–18 h. PMNs were pretreated with human IFNβ (2000 units/ml; PBL Assay Science) for 2 h followed by THG (1 μM) for another 16–18 h before suppression assay as described previously.

**In vivo treatment.** *Depletion of $CD8^+T$ cells.* To deplete $CD8^+T$ cells, 100 μg anti-CD8 (Bio-XCell) or control PBS per mouse was delivered by i.p. injection at day −1, 0, 4, 8, 12, 16, 18, 22. MC38 ($1 \times 10^6$) cells were injected s.c. into C57BL/6 mice. On day 12, TB mice were treated by gavage with p38 inhibitor LY2228829 (1 mg/kg; Selleckchem) prepared in methylcellulose. MC38 TB mice received p38 inhibitor every other day and gavage with analogous volumes of pure methylcellulose was used as vehicle control. Poly I:C (Sigma; 100 μg/ml in PBS, 100 μl/mouse) was injected i.p. every day starting from day 12. In other experiments, MC38 TB mice were treated with 100 μg anti-CD8 followed by Poly I:C and p38i.

**Quantitative real-time PCR.** Total RNA was extracted using the Quick-RNA Microprep Kit (Zymo Research) according to the manufacturer's protocol. cDNA was generated with a High-Capacity cDNA Reverse Transcription Kit (Applied Biosystems). qRT-PCR was performed using Power SYBR Green PCR Master Mix (Applied Biosystems) in 96-well plates. Plates were read with ABI 7500 Fast Real-Time PCR system (Applied Biosystems). Primers are described in Supplementary Table 2.

**RNA-sequencing.** Totally, 75 ng of DNAse I treated, total RNA extracted from PMN-MDSC using the Direct-zol RNA Miniprep (Zymo Research, Irvine, CA) was used to prepare library using the Quant-Seq 3′mRNA-Seq Library Preparation Kit (Lexogen, Vienna, Austria). RNA quantity was determined using the Qubit 2.0 Fluorometer (ThermoFisher Scientific, Waltham, MA) and the quality was validated using the TapeStation RNA ScreenTape (Agilent, Santa Clara, CA). Library quantity was determined using qPCR (KAPA Biosystem, Wilmington, MA) and size was determined using the Agilent TapeStation and the DNA High Sensitivity D5000 ScreenTape (Agilent, Santa Clara, CA). Equimolar amounts of each sample library were pooled, denatured, and High-Output, Single-read, 75 bp cycle. Next-generation sequencing was done on a NextSeq 500 (Illumina, San Diego, CA).

Microarray data from previous study[26] were tested for differences between PMN and PMN-MDSCs using *t* test and *p* values were adjusted for multiple testing by Storey et al. procedure[78]. A list of 633 genes known to be induced by type I interferon at least tenfold in mice was derived from the interferome.org database. Enrichment of interferon-inducible genes among genes significantly downregulated in PMN-MDSC with FDR < 20% was tested using Fisher Exact Test.

RNA-seq data were aligned using Bowtie2 against hg19 genome and RSEM v1.2.12 software[79] was used to estimate gene-level read counts using Ensemble transcriptome information. DESeq2[80] was used to estimate the significance of differential expression difference between the two experimental groups. Gene set enrichment analysis was done on genes passing nominal $p < 0.05$ significance threshold using QIAGEN's Ingenuity® Pathway Analysis software (IPA®, QIAGEN Redwood City, www.qiagen.com/ingenuity) using "Canonical Pathways" options. Only pathways that passed FDR < 10% threshold and had significantly predicted activation state (|Z-score| > 2) were reported.

**ROS production.** Reactive oxygen species were measured by using oxidation-sensitive dye DCFDA (C6827, Molecular Probes/Invitrogen Life Technologies). Splenocytes were labeled with surface markers, washed, and incubated at 37 °C in the serum-free RPMI media in the presence of 3 μM DCFDA. After incubation for 30 min, the cells were analyzed using flow cytometry.

**Detection of ubiquitinated INFAR1 in THP-1 cells.** Totally, $1 \times 10^7$ THP-1 cells were pretreated with vehicle (DMSO) or p38 inhibitor LY2228820 (2 μM, Selleckchem) 1 h prior to 30 min treatment with MC38 tumor conditioned media (TCM) (75%, v/v) or 1 h treatment with Thapsigargin (1 μM, Sigma). Serum-free medium (SFM) served as a negative control to MC38 TCM. Whole-cell lysates were obtained by using 1% NP40 Tris-HCl lysis buffer supplemented with 0.5 M NaF (Sigma), 100 mM $Na_3VO_4$ (Sigma), 1 M β-glycerol phosphate (Sigma), and 10 mM N-Ethylmaleimide (Sigma). Totally, 1.5 mg of whole-cell lysates were mixed with

1.5 μg of anti-IFNAR1 antibody (clone EA12[81]) and incubated in lysis buffer in a final volume of 500 μL at 4 °C under rotation for 4 h. Protein G agarose beads (Invitrogen) were then incubated with whole-cell lysates overnight at 4 °C with rotation. After washing the beads three times with lysis buffer, the proteins were eluted from the beads in a 4× SDS sample loading buffer. Anti-Ub antibodies (clone FK2, Sigma) were used to detect ubiquitinated IFNAR1 protein. Totally, 100 μg of cell lysates were processed to determine p-IFNAR1 (anti-phosphoSer-IFNAR1 antibody[82]) and 60 μg of cell lysates were processed for IFNAR1 (anti-human IFNAR1 antibody, Abcam) and β-actin (clone AC-15, Sigma).

**Statistics.** Statistical analysis was performed using a two-tailed Student's *t* test or Mann–Whitney test after the analysis of the distribution of variables. If the analysis included more than two groups, an ANOVA test with corrections for multiple comparisons was used. Significance was determined at $p < 0.05$. Tumor size evaluation was performed using two-way ANOVA with adjustments for repeated measurements. All calculations were made using GraphPad Prism 8.4.2 software (GraphPad Software Inc.).

**Reporting summary.** Further information on research design is available in the Nature Research Reporting Summary linked to this article.

## Data availability
RNAseq data generated in this study have been deposited in the GEO GenBank under accession code GSE166770. The remaining data are available within the Article, Supplementary Information or available from the authors upon request.

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

## Acknowledgements

We thank members of Koumenis, Minn, Chou, and Hu labs for discussion, Dr. R. Vonderheide for providing cells from KPC mice. We also thank our mentees, Abigail Leighton, Jade Balcombe, and Valerie Irizarry-Negrón for their help. This work was supported by NIH/NCI grants 1R01CA216936 (to D.I.G., Y.N., and S.Y.F.), CA092900 (to S.Y.F.), training grant T32 CA009171 and 1R01CA216936S1 (to K.A.T.), and Wistar Cancer Center Support NIH grant P50 CA168536. ATW and GMA were also supported by R01CA174746 and P01CA114046.

## Authors contibutions

K.A.-T. and E.S. conducted the experiments, analyzed the data, wrote the paper, J.G., F.V., Q. Yu, L.D., C.L., and S.F. conducted the experiments, A.K. analyzed the data, Y.N. provided the material, wrote the paper. C.M., B.N., G.M., F.D., J.B., N.H., A.R.-A. provided material, S.Y.F. and D.I.G. designed research studies, analyzed data, wrote the paper.

## Competing interests

E.S. and D.I.G. are full-time employees of AstraZeneca. The remaining authors declare no competing interests.
