## [Peer Review File · Nature Communications]

Parts of this Peer Review File have been redacted as indicated to remove third-party material where no permission to publish could be obtained.

REVIEWER COMMENTS

Reviewer #1 (Remarks to the Author):

The manuscript by Alicea-Torres and Colleagues aims to determine the restricting role of the signaling through the type I interferons (IFN1) receptor in the functionality of MDSC in tumor-bearing hosts. Authors show the downregulation of the IFN-Alpha R1 (IFNAR1) and IFN1R-related signaling in MDSC from cancer patients and mice with tumors, which depended on the activation of p38 kinase. Notably, loss of IFNAR1 was not sufficient to convert PMN or Mon to MDSC. However, stabilization of IFNAR1 through BM chimeric *Ifnar1*^{S526A} mice, conditioning with IFN β , or promotion of IFNAR1 signaling using a combination of an inhibitor of p38 with IFN-inducing Poly:IC completely ablated the suppressive activity of MDSC and triggered potent antitumor effects mediated by CD8 T cells. The results are interesting, therapeutically significant, and support the emerging role of IFN1-related signatures in the inhibition of MDSC in tumors. Enthusiasm towards the current version of the manuscript was dampened by the unclear overall innovation, the inconsistent presentation of endpoints in unmatching tumor models and MDSC subsets, the incomplete mechanistic connections between ER stress and phospho-p38 in the downregulation of IFNAR1, the undetermined specific role of p38 in the ubiquitination and degradation of IFNAR1 protein in MDSC, and the lack of experiments testing off-target effects of the p38 inhibitor in mice. It is the opinion of the reviewer that addressing these concerns would improve the quality and clarity of the findings. Comments are as follows:

Conceptual Major Points

- 1) The overall innovation of the results is a potential concern. Although the reported downregulation of IFNAR1 in tumor-MDSC and the role of p38 on this phenomenon are novel, several recent reports have elucidated the restricting roles of type I IFN in the functionality of MDSC from tumor-bearing hosts. Thus, the level of innovation of this report is a potential issue.
- 2) There are significant inconsistencies in the models used throughout the paper, which created some confusion. First, the authors showed in some tumor models the readout findings in splenic MDSC, but not in tumor-MDSC, whereas in other tumor models, they showed only tumor-MDSC. Second, authors showed in some experiments results exclusively in PMN-MDSC, whereas in others they indicated effects in both subsets. For consistency and clarity, it is recommended that the effects are shown in splenic and tumor MDSC subsets from the same tumor-models, along with the controls from tumor-free mice.
- 3) Elucidation of the role of p38 in the argued ubiquitination and degradation of IFNAR1 protein in MDSC, rather than a potential effect in IFNAR1 transcriptional control, should be determined. Evaluation of IFNAR1 protein stability, as well as ubiquitination-focused experiments would enable to test these key aspects. Identification of the effector molecules driving ubiquitination and degradation and controlled by p38 would also be important.
- 4) The detailed mechanistic insights whereby the activation of ER stress to p38 restricts the expression of IFNAR1 remain unknown. Identification of the mediators controlling the potential interaction between hypoxia-induced ER stress and p38 in the regulation of IFNAR1 expression would increase the impact of the results.
- 5) It is unclear why PMN-MDSC undergo higher downregulation of IFNAR1 than M-MDSC. Is the p38 role equally effective in both subsets?
- 6) The overall decrease of IFNAR1 between M-MDSC and PMN-MDSC vs. counterparts from tumor-free controls is marginal rather than dramatic.
- 7) Experiments ruling out the off target effects of the combination of Poly:IC (inducer of IFN1) and

LY2228820 in TB mice lacking IFNAR1 in MDSC (perhaps in *Ifnar1*^{S526A} mice) will increase the clarity of this informative therapeutic combination.

Major points on the figures

8) Figure 1:

- A. Figure 1C and 1D: As highlighted above. It is recommended to add results using tumor-MDSC.
- B. Figure 1G: Why do the investigators tested *Irf7* levels in PMN-MDSCs from spleen of Ret Melanoma and KPC mice and in M-MDSC from EL4 tumor-bearing mice. Evaluation of *Irf7* should be compared in both subsets from spleen and tumor in the same tumor models.
- C. Figure 1H: It is unclear whether alterations in *Isg15* are also found in M-MDSC as reported for EL4-associated PMN-MDSC.

9) Figure 2:

- A. In the MDSC suppression assay, some experiments are showing % of proliferation (that exceeds 100 % in figure 3D and 3E) and others are showing CPM. It is unclear why authors used different ways to illustrate the effects in proliferation.
- B. Figure 2B: Does supplementation with IFN β also abrogate the immunosuppressive activity of M-MDSC as it is illustrated for PMN-MDSC?

10) Figure 3: The restoration of tumor growth in SA TB mice after CD8 depletion is interesting. The addition of readouts indicating effector-cytotoxic functionality of tumor-infiltrating T cells will complement and increase the impact of the findings.

11) Figure 6: It is unclear whether exposure to tumor explants also induces phosphorylation of p38 in M-MDSC. Identification of the tumor microenvironment factors driving the phosphorylation of p38 would be highly impactful and beneficial for the overall message.

12) Figure 7:

- A. Figure 7A and 7B. The condition using p38 inhibitor alone needs to be included.
- B. Figure 7G. The addition of experiments showing that the combination of p38 inhibitor plus poly IC promotes the expansion of tumor-specific T cells would be relevant.

Minor comments:

1. In page 4: Patients instead of patents.
2. In figure 1H: Are the results presented in fold changes or relative expression?
3. Figure 7C and 7D are switched in the narrative.

Reviewer #2 (Remarks to the Author):

Summary of the main findings

The process leading to generation of functionally active myeloid-derived suppressor cells (MDSCs) involves different steps and signals. Authors present here data in support for the down-regulation of type I interferon (IFN1) receptor signaling, mostly in neutrophils but also in monocytes, as a prerequisite for the acquisition of the immune suppressive activity. Downregulation of the IFNAR1 component of the type I IFN receptor and the ensuing inactivation of the IFN1 pathway was related to the activity of p38 protein kinase. From a therapeutic perspective, the combination between a small molecule inhibitor targeting 38 with the interferon-inducing administration of Poly:IC resulted in synergistic, anti-tumor activity in a mouse model.

Main issues

The manuscript leaves some questions unanswered.

What factors in the TES are activating p38 and affecting IFNAR recycling?

What is the IFN β contribution to the inhibition of suppressive activity induced by THG? How is the cytokine addition linked to the IFNAR1 recycling and signalling? THG might have different activities on the cells; to prove that IFNAR1 is the main target, additional experiments are needed. Why p38 inhibitors only work in combination with strong IFN inducers like Poly IC? These results are not easily explained considering the anti-tumor phenotype of SA mice.

Specific issues

- 1) There is a random use of the data reporting in different figures. For example, the suppression by different cells are shown as either percentage of proliferation or CPM, either in bars or dots.
- 2) In Fig S1A, the signature referring to type I IFNAR1 looks like a broader immune downregulation, which could involve also IFN- γ -dependent genes among other pathways.
- 3) Fig. 3B. The controls where SA mice are grafted with bone marrow from both sources are missing.
- 4) Fig. 3D, right panel. In this panel, I would think that the PMN-MDSCs in the tumor are suppressive, and the only difference is visible at 1:4 cell ratio.
- 5) Fig. 3G. This panel is not easy to follow unless the reader retrieves the information by swapping among the text and the legends. Moreover, it is difficult to think that the groups WT and SA in tumor are not significantly different even if there are only three points.
- 6) It is problematic to see an effect of hypoxia in Fig. 6E. It looks like the genetic ablation of p38 is the dominant factor for the IFNAR1 expression. Moreover, in the panel F of the same figure, the effect of different TES on p38 phosphorylation are quite variable and no real explanation is offered. Finally, it should be interesting to relate them directly to the action on IFNAR1 expression. It is stated that the Fig. 5C is related to this experiments but the TES are reported with different labels, making quite hard to compare the two graphs side by side.

Reviewer #3 (Remarks to the Author):

The authors of this work showed that Type 1 INF receptor signaling in neutrophils and monocytes serves as a universal mechanism that restricts acquisition of suppressive activity by MDSCs. They demonstrate that downregulation of the IFNAR1 chain of this receptor and inactivation of the INF1 pathway is present in MDSCs from cancer patients and several mouse tumor models and that decreased receptor presence depends on the activation of the p38 protein kinase and promotes the immune suppressive phenotype. They also establish that genetic stabilization of IFNAR1 in tumor bearing mice undermines the suppressive activity of MDSC and results in antitumor responses. Furthermore, they show that stabilizing the INF receptor using p38 inhibitor combined with the INF induction therapy using Poly:IC, results in strong anti-tumor responses. These findings are overall interesting and potentially important. However, the studies are limited only on one subunit of the receptor, while the potential involvement of IFNAR2 was not explored, somewhat limiting the impact of this work.

SPECIFIC ISSUES

- a. The authors demonstrate convincingly in Figure 1A that PMN-MDSC demonstrate markedly lower expression of IFNAR1 than PMN from healthy donors. It is unclear why the authors only investigated expression of IFNAR1 and not IFNAR2 also. Is the downregulation of IFNAR1 selective or it also involves IFNAR2. This is relevant as different Jak kinases bind to the different receptor subunits and elements of the pathway could be potentially preserved via IFNAR2.
- b. The studies in figure 2 show that deletion of *ifnar1* is not sufficient to convert PMN or Mon to MDSC. Is that the case in mice *ifnar2* knockout mice? If someone is to draw conclusions on the role of the INF receptor, studies with *ifnar2*^{-/-} mice would be necessary, especially as there is previous evidence for distinct biological differences between *ifnar1*^{-/-} and *ifnar2*^{-/-} mice (Shepardson et al, Front Immunol 2018).

- c. Figure 6, panel B. Have the authors examined effects on other Map kinases? Was phospho-Erk and phospho-JNK also inducible?
- d. Figure 7, panel A & B. The authors demonstrate that the p38 inhibitor ralimetinib abrogates the downregulation of the IFNAR1. What is the effect of the p38 inhibitor on the IFN-induced phosphorylation of Tyk2 that is associated with IFNAR1? What is the effect on phosphorylation of effector Stat proteins?
- e. It would be interesting to determine the effects of p38 inhibition on IFNAR2. Is there a similar mechanism involving that subunit of the receptor and how does this impact downstream signaling and resulting biological activities?

We thank our reviewers for their thoughtful comments. In response we performed large number of experiments and made changes in the text and figures to address all their questions and concerns. We included 11 new figures to this response. Specific point-by-point responses are provided below.

Reviewer # 1.

Conceptual Major Points

1) The overall innovation of the results is a potential concern. Although the reported downregulation of IFNAR1 in tumor-MDSC and the role of p38 on this phenomenon are novel, several recent reports have elucidated the restricting roles of type I IFN in the functionality of MDSC from tumor-bearing hosts. Thus, the level of innovation of this report is a potential issue.

We appreciate this important point. As we have noted, the literature on the role of IFN1 in control of MDSC is rather controversial. Reports suggesting that IFN1 stimulate the generation¹ or suppressive activities of MDSC via sustaining expression of PD-L1² are contradicted by the studies where administration of IFN1 inducers such as poly(I:C) or CpG led to decreased numbers or/and undermined suppressive activities of MDSC³⁻⁵. Whereas activation of STING and ensuing production of IFN1 by tumor irradiation was suggested to induce MDSC⁶, the opposite results were reported by studies using forced expression of STING; however, the latter effects were not dependent on IFN1⁷. Thus, the critical questions regarding the possible role of IFN1 in regulation of MDSC suppressive activity as well as the mechanisms by which the effects of IFN1 on MDSC are inactivated in the tumor microenvironment remains to be answered. In the revised manuscript, we have further emphasized both potential importance of IFN1 in the regulation of MDSC and a substantial gap in our knowledge regarding these mechanisms.

The novelty of our study (mentioned by our reviewer) is in a definitive demonstration that IFN1 acts to inactivate the suppressive properties of MDSC and this novel mechanism of negative regulation of MDSC is being undermined by downregulation of IFNAR1. Moreover, we found that inhibition of p38 increased IFNAR1 expression on PMN-MDSC and rendered them responsive to type 1 IFN. We believe that these data are novel, important and will be of interest to the broad readership of *Nature Communications*.

2) There are significant inconsistencies in the models used throughout the paper, which created some confusion. First, the authors showed in some tumor models the readout findings in splenic MDSC, but not in tumor-MDSC, whereas in other tumor models, they showed only tumor-MDSC. Second, authors showed in some experiments results exclusively in PMN-MDSC, whereas in others they indicated effects in both subsets. For consistency and clarity, it is recommended that the effects are shown in splenic and tumor MDSC subsets from the same tumor-models, along with the controls from tumor-free mice.

We thanks our reviewer for this comment. The first version of the paper presented some inconsistencies in IFNAR1 quantification on MDSC subsets in different tumor models. This was due to technical issues with receptor degradation caused by digesting enzymes commonly used to isolate tumor-infiltrating leukocytes. We now have adopted method of isolation of intratumoral leukocytes by mechanical de-aggregation of the tumors. We added data showing

IFNAR1 levels on splenic and tumor-infiltrating PMN-MDSC and M-MDSC from 3 subcutaneous tumor models and one transgenic model (KPC). In all models analyzed, substantial down-regulation of IFNAR1 expression was observed on PMN-MDSC. IFNAR1 downregulation was more prominent in the tumors than in spleens. M-MDSC downregulated IFNAR1 expression was not detected (**Figure for Review 1**). The results are included as **Fig. 1F** to the new version of the manuscript.

Figure for Review 1. Expression of IFNAR1 on MDSC in tumor-bearing mice. IFNAR1 expression on was measured by flow cytometry on splenic or the surface of indicated cells isolated from spleens (SPL) or from tumors (TUM) of tumor-bearing mice. Cells were obtained by mechanical de-aggregation of subcutaneous LLC, B16-F10 or CT26 tumors. Each dot represent a single mouse. Mean and SD of geometric MFI are shown. P values were calculated in two-sided t-test with. * - $p < 0.05$; ** - $p < 0.01$; ***- $p < 0.001$

3) *Elucidation of the role of p38 in the argued ubiquitination and degradation of IFNAR1 protein in MDSC, rather than a potential effect in IFNAR1 transcriptional control, should be determined. Evaluation of IFNAR1 protein stability, as well as ubiquitination-focused*

experiments would enable to test these key aspects. Identification of the effector molecules driving ubiquitination and degradation and controlled by p38 would also be important.

This is an excellent point. We have analyzed the levels of IFNAR1 mRNA in MDSC and found no difference suggesting that a post-translational mode of IFNAR1 regulation (e.g. protein degradation). This point is further supported by the lack of IFNAR1 downregulation in MDSC from SA mice, whose receptor lacks a key Serine residue needed for ubiquitination.

Low levels of IFNAR1 expression and lack of sufficient material for biochemical studies make it difficult to evaluate endogenous IFNAR1 ubiquitination and degradation in MDSC. Therefore, we performed experiments with THP-1 monocytic cell line derived from patient with monocytic leukemia. ER stress inducer Thapsigargin (THG) (**Figure for Review 2A**) and tumor cell conditioned medium (**Figure for Review 2B**) caused phosphorylation of IFNAR1 and decrease in IFNAR1 protein. This was associated with IFNAR1 ubiquitination (**Figure for Review 2**). P38 inhibitor LY2228820 abrogated those effects of tumor cells conditioned medium and THG (**Figure for Review 2**).

Figure for Review 2. Ubiquitination and degradation of IFNAR1 in monocytic cell line. 1×10^7 of THP-1 cells were pretreated with vehicle (DMSO) or p38 inhibitor LY2228820 1h prior to treatment with 1h of Thapsigargin (THG) (1 μ M, Sigma) (A) or 30 min of MC38 tumor conditioned media (75%, v/v) or Serum free medium (SFM) (B). Cell lysates were immunoprecipitated with IFNAR1 antibody and probed with anti-ubiquitin antibody. Whole cell lysates (WCL) were used to evaluate IFNAR1 and pIFNAR1 proteins. Two experiments with the same results were performed.

These results were consistent with the role of the p38 kinase in the phosphorylation-dependent ubiquitination and degradation of IFNAR1 in other cell systems. Downregulation of IFNAR1 within tumor microenvironment mediated by yet to be identified soluble factors⁸ and extracellular vesicles⁹ produced by cancer cells rely on the coordinated activity of casein kinase 1 α and p38 α stress activated protein kinase¹⁰⁻¹³ to drive the phosphorylation-dependent ubiquitination and degradation of IFNAR1. This phosphorylation and ensuing IFNAR1 ubiquitination can be induced by IFN1 (in a manner dependent on Janus kinases and protein kinase D2^{14,15}), or by a large group of the non-ligand stimuli (such as ER stress^{16,17}, integrated stress response induced by hypoxia or deficit of amino acids¹⁸, pathogen recognition receptors¹⁹, products of tobacco smoking²⁰, and pro-inflammatory cytokines^{21,22}).

We think that our data together with available evidence in the literature is sufficient to propose that IFNAR1 downregulation in MDSC is due to the downregulation and degradation of this receptor.

4) *The detailed mechanistic insights whereby the activation of ER stress to p38 restricts the expression of IFNAR1 remain unknown. Identification of the mediators controlling the potential interaction between hypoxia-induced ER stress and p38 in the regulation of IFNAR1 expression would increase the impact of the results.*

As stated in the previous point, the mechanisms by which p38 (activated by the ER stress or other relevant to tumor microenvironment inducers) stimulates downregulation of IFNAR1 have been explored and described in our previous publications^{8-11,19,21}. Specific mediators depend on the nature of a stimulus. For example (data from Ref¹⁸ presented below), the activation of p38 and downregulation of IFNAR1 in response to thapsigargin or hypoxia requires PERK whereas the same events stimulated by the lack of amino acids are dependent on GCN2 kinase. Hypoxia downregulates IFNAR1 in human melanoma cells in a PERK-dependent manner (RNAi used here)

[Redacted]

In mouse embryo fibroblasts of indicated genotypes ablation of either PERK or GCN2 selectively affect p38 activation, IFNAR1 phosphorylation and, at a latter time point (lower panel), IFNAR1 downregulation induced either by thapsigargin (TG) or leucine starvation (LS) but not by PKR activator Heparin Sulfate (HE)

Given previously published information, in this work, we focused on the functional importance of inactivation of IFNAR1 for MDSCs for their immune suppressive activities.

5) *It is unclear why PMN-MDSC undergo higher downregulation of IFNAR1 than M-MDSC. Is the p38 role equally effective in both subsets?*

We performed additional experiment where we compared side-by-side the amount of phosphorylated p38 in monocytes and neutrophils. Monocytes had higher basal level of p-p38 than neutrophils and it is only slightly increased by TES treatment as compared to neutrophils where TES caused substantial up-regulation of p-p38 (**Figure for Review 3**). This result was included to manuscript as **Fig. 6F**

Figure for Review 3. Effect of tumor explant supernatants on amount of phosphorylated p38 in monocytes and neutrophils. Monocytes and PMN were isolated from bone marrows from naïve mice, treated for 18h in vitro with 20% of TES, cells were then lysed and phospho-p38 and total p38 measured by western blot. Experiments were repeated once with the same result.

6) *The overall decrease of IFNAR1 between M-MDSC and PMN-MDSC vs. counterparts from tumor-free controls is marginal rather than dramatic.*

The decreased in IFNAR1 in PMN-MDSC was highly significant in all models and in cancer patients. In several models expression in PMN-MDSC was 3-fold lower than in control neutrophils. M-MDSC showed smaller effect and discussed it in the manuscript

7) *Experiments ruling out the off target effects of the combination of Poly:IC (inducer of IFN1) and LY2228820 in TB mice lacking IFNAR1 in MDSC (perhaps in *Ifnar1*^{S526A} mice) will increase the clarity of this informative therapeutic combination.*

We thank the reviewer for the comment. We performed the required experiments and we think the results strengthen the overall message of the paper. We utilized mice that have depletion of IFNAR1 in neutrophils and monocytes (*IFNAR1*^{fl/fl}*S100A8*^{Cre}). We observed no anti-tumor effect of poly-I:C and LY2228820 treatment in those mice (**Figure for Review 4**). The results are included as **Fig. 7I** to new version of the manuscript. These results indicate that effect of p38 inhibitor was mediated in large part by IFNAR1 expression on MDSC.

Major points on the figures

8) *Figure 1:*

A. *Figure 1C and 1D: As highlighted above. It is recommended to add results using tumor-MDSC.*

We analyzed IFNAR1 level on PMN and M-MDSC in 4 different models including LLC, B16, and CT26 comparing them with naïve counterparts (**Figure for Review 1**)

Figure for Review 4. Effect of p38 inhibitor and poly:IC treatment on tumor growth. 10^6 MC38 cells were injected in the right flank of the mice lacking IFNAR1 in myeloid cells (IFNAR1^{fl/fl}Cre⁺ mice) and tumor growth was measured every 2-4 days (n=4-5 per group). Treatment started at day 12 with poly:IC in PBS (10 μ g/mouse) i.p. daily and p38 inhibitor (LY2228829 1 mg/kg) prepared in methylcellulose administered by oral gavage every other day. Control mice received equal amount of vehicle (PBS i.p. and methylcellulose by oral gavage). P values are calculated by two-way ANOVA test with correction for repeated measurements.

B. Figure 1G: Why do the investigators tested Irf7 levels in PMN-MDSCs from spleen of Ret Melanoma and KPC mice and in M-MDSC from EL4 tumor-bearing mice. Evaluation of Irf7 should be compared in both subsets from spleen and tumor in the same tumor models.

We thanks the reviewer for the suggestion. We now have analyzed Irf7 and Isg15 from splenic PMN and monocytes and MDSC from tumor-bearing mice. Results are presented in **Figure for Review 5** below and in **Fig. 2A** in the manuscript.

C. Figure 1H: It is unclear whether alterations in Isg15 are also found in M-MDSC as reported for EL4-associated PMN-MDSC.

We added these results and presented then in **Figure for Review 5** and **Fig. 2A** in the manuscript.

Figure for Review 5. Expression of interferon response genes in MDSC. PMN and Monocytes were isolated by cell sorting from the spleen of naïve mice (Naïve SP) or spleen (TUM SP) and tumors (TUM) of EL4-bearing mice. Irf7 and Isg15 expression was quantified by qRT-PCR in each cell population. Each dot represent a single mouse. Mean and SD are shown (n=6). P values were calculated in ANOVA test with corrections for multiple comparisons.

9) *Figure 2:*

A. *In the MDSC suppression assay, some experiments are showing % of proliferation (that exceeds 100 % in figure 3D and 3E) and others are showing CPM. It is unclear why authors used different ways to illustrate the effects in proliferation.*

If experiments were performed at the same time we used direct value of CPM. However, if the experiments were performed at different time (it is difficult to obtain enough cells from tumors to performed multiple biological replicates in one experiment) the direct comparison of CPM values could be rather difficult due to variability of background/basal values. In order to evaluate biological replicates in those cases we used % of changes against background rather than actual CPM.

B. *Figure 2B: Does supplementation with IFN β also abrogate the immunosuppressive activity of M-MDSC as it is illustrated for PMN-MDSC?*

We performed the experiments by treating M-MDSC with IFN β . The treatment reversed the suppressive activity of M-MDSC. Results are presented in **Figure for Review 6** and in **Fig. 2C** in the manuscript.

Figure for Review 6. Effect of IFN β on suppressive activity of M-MDSC. M-MDSC were isolated from the spleen of EL4-bearing mice, pre-treated for 2h with IFN β (2000U/ml). IFN β was then washed away and the suppressive activity of the cells was measured by co-culturing M-MDSC with PMEL splenocytes stained with Cell trace Far red at different M-MDSC: splenocytes ratio stimulated with the cognate peptide (gp100 10ng/ml). Cell trace dilution was measured after 48h to assess proliferation of CD8 $^+$ T cells. Mean and SD are shown (n=3). P values were calculated in two-sided Student's t-test.

10) *Figure 3: The restoration of tumor growth in SA TB mice after CD8 depletion is interesting. The addition of readouts indicating effector-cytotoxic functionality of tumor-infiltrating T cells will complement and increase the impact of the findings.*

Characterization of T cell function in SA mice was described in our previous publication²³ and additional studies on the status of cytotoxic lymphocytes would require much deeper molecular analysis and are outside the scope of this manuscript.

11) *Figure 6: It is unclear whether exposure to tumor explants also induces phosphorylation of p38 in M-MDSC. Identification of the tumor microenvironment factors driving the phosphorylation of p38 would be highly impactful and beneficial for the overall message.*

We performed a new experiment using PMN and Monocytes treated with TES to assess their effect of p38 phosphorylation. The results are presented and discussed in **Figure for Review 3**.

12) *Figure 7:*

A. *Figure 7A and 7B. The condition using p38 inhibitor alone needs to be included.*

We thank our reviewer for the comment. We performed new experiments using healthy donor and murine PMN and monocytes treated with p38 inhibitor alone. P38 inhibitor caused up-regulation of IFNAR1 with and without the presence of TES. These results are shown in **Figure for Review 7** and included to **Fig. 7A, B** in the manuscript.

Figure for Review 7. The effect of p38 inhibitor on expression of IFNAR1 in neutrophils and monocytes. (A) PMN were isolated from blood of healthy volunteers and cultured for 16-18h with medium containing 20 ng/ml of GM-CSF with or without p38i (LY2228829 1μM) and IFNAR1 level was measured by flow cytometry. Each dot represent one HD. P value (*-p<0.05) was calculated in two-sided Student's t-test. (B) Mon (left panel) and PMN (right panel) were isolated from bone marrow of mice by cell sorting, pre-treated with 1 μM p38i followed by incubation with 20% TES (n=3) for 16-18h. IFNAR1 was measured by flow cytometry. Geometric MFI is shown. P values were calculated in ANOVA test with corrections for multiple comparisons.

B. *Figure 7G. The addition of experiments showing that the combination of p38 inhibitor plus poly IC promotes the expansion of tumor-specific T cells would be relevant.*

It is overall a good idea and warrants further investigation. However these additional studies on the status of cytotoxic lymphocytes are currently outside the scope of this manuscript.

Minor comments:

1. *In page 4: Patients instead of patents.*

This error was corrected

2. *In figure 1H: Are the results presented in fold changes or relative expression?*

This is a new Figure 2A and data are now presented as relative expression

3. *Figure 7C and 7D are switched in the narrative.*

This mistake was corrected

Reviewer #2

Main issues

What factors in the TES are activating p38 and affecting IFNAR recycling?

Our previous work demonstrated that a number of factors that could be present in TES are capable of activating p38 leading to ubiquitination and downregulation of IFNAR1. These factors include TNF α , IL-1 α , IL-1 β (as reported in Ref ²¹) as well as inducers of the pathogen recognition receptors (various DAMPs and PAMPs – as in Ref ¹⁹), some additional yet to be identified soluble factors⁸ as well as extracellular vesicles⁹ produced by cancer cells). One example (data from Ref ⁹) is provided below.

[Redacted]

Treatment of WT splenocytes with tumor-derived extracellular vehicles (TEV) downregulates IFNAR1. This phenotype is not seen when p38 α (Mapk14) is ablated or in SA mice whose IFNAR1 cannot be phosphorylated and ubiquitination. The same work also contains data on the effects of TEV on p38 activation.

What is the IFNbeta contribution to the inhibition of suppressive activity induced by THG? How is the cytokine addition linked to the IFNAR1 recycling and signalling?

As outlined in our response to Reviewer 1, IFNAR1 can get phosphorylated, ubiquitinated and downregulated either by its ligands (i.e. IFN β) or by the non-ligand inducers (thapsigargin, TEV, hypoxia, inflammatory cytokines, etc). If cells first encounter IFN, they will initiate IFN response and then also downregulate IFNAR1. Such cells will express the IFN-stimulated genes but be less sensitive to the next encounter with IFN. If cells first encounter a non-ligand inducer, they will lose IFNAR1 without any benefit of IFN signaling and responses. Such cells will be unresponsive to IFN and will robustly induce the IFN-stimulated genes. This has been documented in several of our primary papers as well as reviews ^{8-11,16-18,21,24-26}.

In this particular case, pre-treatment with IFN β is used to initiate the IFN pathway prior to IFNAR1 downregulation in response to thapsigargin – so the cells have a chance to respond. Data indicating that, in these settings, IFN β inactivate the suppressive properties of MDSCs supports the notion that type I IFN is a negative regulator of MDSCs activities – for as long as it has a receptor to act upon.

Consistent with the results described above, in our experimental system treatment of PMN with IFN β caused down-regulation of IFNAR1 with associated up-regulation of *Irf7* and *Isg15* (**Figure for Review 8**).

THG might have different activities on the cells; to prove that IFNAR1 is the main target, additional experiments are needed.

This is important question that we addressed with additional experiments. As we described in manuscript, THG induced immune suppressive activity by PMN. We tested the hypothesis that IFN1 signaling will be sufficient to neutralize the effect of THG. THG cause similar level of suppressive activity in PMN from WT and SA mice. Pre-treatment of PMN with INF β did not

Figure for Review 8. Effect of IFN β on IFNAR1 and interferon response gens in PMN. PMN were isolated from bone marrow of naïve mice, treated *in vitro* with 2000U/ml of IFN β for 2h, then IFNAR1 was measured by flow cytometry. *Irf7* and *Isg15* expression was measured by qRT-PCR in the same cells. (n=3). P values were calculated in two-sided Student's t-test.

prevent induction of suppressive activity by THG. (**Figure for Review 9**). Thus, at least in experiments *in vitro*, THG caused effect that was not abrogated by IFN1 signaling. Whether this is phenomenon associated with strong induction of ER stress or other mechanisms is a subject of further investigations. The results are included to the manuscript as **Fig. 5G,H**.

Figure for review 9. Effect of IFN1 signaling on THG-induced suppressive activity. (A) PMN were isolated from BM of naïve WT or SA mice, treated overnight with 1 μ M of THG. THG was then washed away and suppressive activity of the cells was measured by co-culturing PMN with PMEL splenocytes stained with Cell trace Far red stimulated with the cognate peptide (gp100, 10ng/ml). Cell trace dilution was measured after 48h to assess proliferation of CD8⁺ T cells (n=3). (B) PMN were isolated from BM of naïve WT or SA mice pre-treated for 2h with IFN β (2000U/ml) then overnight with 1 μ M of THG. Suppressive activity of the cells was measured by co-culturing PMN with PMEL splenocytes stained with Cell trace Far red stimulated with the cognate peptide (gp100, 10ng/ml). Cell trace dilution was measured after 48h to assess proliferation of CD8⁺ T cells.

Why p38 inhibitors only work in combination with strong IFN inducers like Poly IC? These results are not easily explained considering the anti-tumor phenotype of SA mice.

PMN and Mon in SA mice because of high IFNAR1 expression level have strong IFN signaling from the beginning, when tumors were implanted and cause initial inflammatory reaction. Therefore, effect can be detected without need to additional source of IFN1. It is important to point out that in SA mice antitumor effect is rather modest and reversed with time. In wild-type mice, tumor growth is associated with accumulation of MDSC with reduced level of IFNAR1. In this case, the presence of IFN1 in tumors and circulation was not sufficient to induce potent

response. P38 inhibitors elicit partial response that could be augmented by additional source of IFN1 (poly:IC).

Specific issues

1) *There is a random use of the data reporting in different figures. For example, the suppression by different cells are shown as either percentage of proliferation or CPM, either in bars or dots.*

As we described in response to Reviewer 1 if experiments were performed at the same time we used direct value of CPM. However, if the experiments were performed at different time (it is difficult to obtain enough cells from tumors to performed multiple biological replicates in one experiment) the direct comparison of CPM values could be rather difficult due to variability of background/basal values. In order to evaluate biological replicates in those cases we used % of changes against background rather than actual CPM.

We revised some figures to provide for consistency.

2) *In Fig S1A, the signature referring to type I IFNAR1 looks like a broader immune downregulation, which could involve also IFN-gamma-dependent genes among other pathways.*

This point is well taken. We have revised text to address this point.

3) *Fig. 3B. The controls where SA mice are grafted with bone marrow from both sources are missing.*

We did not perform this experiment in the current study because they have been done previously. These studies (published in Ref. ^{22,23,27}) demonstrate that the pro-inflammatory and anti-tumorigenic phenotypes of non-degradable IFNAR1^{SA} are largely mediated by the bone marrow cells.

4) *Fig. 3D, right panel. In this panel, I would think that the PMN-MDSCs in the tumor are suppressive, and the only difference is visible at 1:4 cell ratio.*

Our reviewer is correct. PMN-MDSC from tumor are more suppressive. However, isolation of these cells from tumors is very tricky due to difficulties to collect cells from tissues after digestion. Therefore, there is a variability in ratios in different experiment.

5) *Fig. 3G. This panel is not easy to follow unless the reader retrieves the information by swapping among the text and the legends. Moreover, it is difficult to think that the groups WT and SA in tumor are not significantly different even if there are only three points.*

We have revised figure and performed different statistical test to determine significance.

6) *It is problematic to see an effect of hypoxia in Fig. 6E. It looks like the genetic ablation of p38 is the dominant factor for the IFNAR1 expression. Moreover, in the panel F of the same figure, the effect of different TES on p38 phosphorylation are quite variable and no real explanation is offered.*

Phosphorylation of p38 is regulated by many non-redundant factors. Hypoxia and soluble factors present in TES are only several of many possible mechanisms present in tumor microenvironment. We have revised discussion to address this issue.

Finally, it should be interesting to relate them directly to the action on IFNAR1 expression. It is stated that the Fig. 5C is related to this experiments but the TES are reported with different labels, making quite hard to compare the two graphs side by side.

We revised the text to improve presentation of the results.

Reviewer #3

SPECIFIC ISSUES

a. The authors demonstrate convincingly in Figure 1A that PMN-MDSC demonstrate markedly lower expression of IFNAR1 than PMN from healthy donors. It is unclear why the authors only investigated expression of IFNAR1 and not IFNAR2 also. Is the downregulation of IFNAR1 selective or it also involves IFNAR2. This is relevant as different Jak kinases bind to the different receptor subunits and elements of the pathway could be potentially preserved via IFNAR2.

We thank our reviewer for the comment and suggestion. We quantified IFNAR2 levels on human PMN after the treatment with TES (that induces IFNAR1 downregulation. IFNAR2 was not downregulated by TES treatment as IFNAR1 suggesting that only IFNAR1 is downregulated by the tumor microenvironment (**Figure for Review 10**). The results are included to manuscript as **Fig. S4A**. Please note we could perform these experiments only in human samples because there is no adequate mouse antibody currently available.

Figure for Review 10. Effect of TES on IFNAR2 expression. PMN were isolated from blood of healthy donor volunteers and cultured for 16-18h with medium containing 20ng/ml of GM-CSF with or without 30% of TES. IFNAR2 level was measured by flow cytometry. Each dot represent individual volunteer. Geometric MFI is shown

b. The studies in figure 2 show that deletion of ifnar1 is not sufficient to convert PMN or Mon to MDSC. Is that the case in mice ifnar2 knockout mice? If someone is to draw conclusions on the role of the INF receptor, studies with ifnar2-/- mice would be necessary, especially as there is previous evidence for distinct biological differences between ifnar1-/- and ifnar2-/- mice (Shepardson et al, Front Immunol 2018).

We thanks the reviewer for this comment and bringing our attention to IFNAR2. We focused on IFNAR1 because current paradigm of type I IFN signaling supported by many published papers indicates that IFN signaling is impossible without IFNAR1. To best of our knowledge, there are

no report that demonstrates signaling of IFN β (or another type I IFN) through IFNAR2 without requiring IFNAR1. Shepardson et al (Front Immunol 2018) did not investigate IFN signaling but reported that, in their hands, *Ifnar2* (but not *Ifnar1*) knockout mice exhibit an influenza phenotype. They loosely quoted deWeerd et al (Nature Immunol 2013) as evidence that there could be an IFNAR2-specific signaling and proposed that such signaling is involved in the animal phenotypes they observe. We have serious issue with this publication. First, a robust influenza phenotype in the *Ifnar1*^{-/-} is reported by many publications (for example,²⁸⁻³³). Second, de Weerd et al (Nature Immunol 2013³⁴) proposed an IFNAR2-independent signaling by IFN β via IFNAR1 alone. In this case, IFNAR1 is absolutely essential for IFN signaling (regardless of the presence of IFNAR2). Third, recent comprehensive and exhaustive CRISPR studies demonstrated that knockout of IFNAR1 completely abrogates all types of IFN signal transduction and expression of the IFN-stimulated genes (Urin et al, J Mol Biol. 2019).

Nevertheless, we believe that question raised by our reviewer was very important and deserved direct experimental testing. In collaboration with Dr. Rynda-Apple, who published this paper (Shepardson et al, Front Immunol 2018³⁵), we procured myeloid cells from the IFNAR2-deficient mice (*Ifnar2*^{tm1(KOMP)Vlclg}), which were originally purchased from UC Davis KOMP Repository. These 6-8 weeks old mice were bred and maintained at Montana State University (Bozeman, MT) Animal Resources Center under pathogen-free conditions in accordance with the recommendations of the NIH and USDA under a protocol approved by the MSU Institutional Animal Care and Use Committee (IACUC). we obtained cells from these IFNAR2 KO mice³⁵ and we performed the experiments to determine the possibility that lack of IFNAR2 alone could drive the conversion of PMN in PMN-MDSC. We performed T-cells suppression experiment using PMN from naive WT and IFNAR2 KO mice. No suppression was detected (**Figure for Review 11**). The results were added to the manuscript as **Fig. S4B**.

Figure for Review 11. Lack of IFNAR2 did not convert PMN to PMN-MDSC. PMN were isolated from BM of naive WT or IFNAR2 KO mice and suppressive activity of the cells was measured by co-culturing PMN with PMEL splenocytes stained with Cell trace Far red at different ratio stimulated with the cognate peptide (gp100 10 ng/ml). Cell trace dilution was measured after 48h to assess proliferation of CD8+ T cells (n=3).

c. Figure 6, panel B. Have the authors examined effects on other Map kinases? Was phospho-Erk and phospho-JNK also inducible?

Previously published results demonstrate that inhibitors of MEK1 and JNK did not affect phosphorylation of IFNAR1¹⁹. Since as we described in our responses above only p38 was found to be relevant for IFNAR1 downregulation, we considered those experiments as superfluous.

d. Figure 7, panel A & B. The authors demonstrate that the p38 inhibitor ralimetinib abrogates

the downregulation of the IFNAR1. What is the effect of the p38 inhibitor on the IFN-induced phosphorylation of Tyk2 that is associated with IFNAR1? What is the effect on phosphorylation of effector Stat proteins?

[Redacted]

As shown in our published work (Ref ¹¹), inactivation of p38 increases STAT phosphorylation and activation further supporting the role of p38 in tempering the extent of IFN signaling. These results show that activation of p38-driven downregulation of IFNAR1 attenuates STAT1 phosphorylation stimulated by IFN β . Importantly, this phenotype can be reversed by knockout or knockdown of p38 or by p38 inhibitor (VX-702).

harvested. Analyses of STAT1 phosphorylation and levels are shown. *D*, analyses of STAT1 levels and phosphorylation in lysates from 2fTGH cells that received indicated shRNA were pretreated or not with TG (1 μ M, 3 h) and then treated with human IFN α (200 international units/ml for 30 min) as indicated. *E*, analyses of STAT1 levels and phosphorylation in lysates from MEFs pretreated or not with TG (1 μ M, 3 h) and then treated with mouse IFN β (50 international units/ml for 30 min) as indicated.

e. It would be interesting to determine the effects of p38 inhibition on IFNAR2. Is there a similar mechanism involving that subunit of the receptor and how does this impact downstream signaling and resulting biological activities?

Since our data presented in Figures for Review 10 and 11 did not involvement of IFNAR2 we did not follow this line of research further.

References used in the response letter

1. Taleb, K., *et al.* Chronic Type I IFN Is Sufficient To Promote Immunosuppression through Accumulation of Myeloid-Derived Suppressor Cells. *J Immunol* **198**, 1156-1163 (2017).
2. Xiao, W., Klement, J.D., Lu, C., Ibrahim, M.L. & Liu, K. IFNAR1 Controls Autocrine Type I IFN Regulation of PD-L1 Expression in Myeloid-Derived Suppressor Cells. *Journal of immunology* **201**, 264-277 (2018).
3. Shirota, Y., Shirota, H. & Klinman, D.M. Intratumoral injection of CpG oligonucleotides induces the differentiation and reduces the immunosuppressive activity of myeloid-derived suppressor cells. *Journal of immunology* **188**, 1592-1599 (2012).

4. Zoglmeier, C., *et al.* CpG blocks immunosuppression by myeloid-derived suppressor cells in tumor-bearing mice. *Clin Cancer Res* **17**, 1765-1775 (2011).
5. Metzger, P., *et al.* Immunostimulatory RNA leads to functional reprogramming of myeloid-derived suppressor cells in pancreatic cancer. *J Immunother Cancer* **7**, 288 (2019).
6. Liang, H., *et al.* Host STING-dependent MDSC mobilization drives extrinsic radiation resistance. *Nat Commun* **8**, 1736 (2017).
7. Zhang, C.X., *et al.* STING signaling remodels the tumor microenvironment by antagonizing myeloid-derived suppressor cell expansion. *Cell Death Differ* **26**, 2314-2328 (2019).
8. HuangFu, W.C., Qian, J., Liu, C., Rui, H. & Fuchs, S.Y. Melanoma cell-secreted soluble factor that stimulates ubiquitination and degradation of the interferon alpha receptor and attenuates its signaling. *Pigment Cell Melanoma Res* **23**, 838-840 (2010).
9. Ortiz, A., *et al.* An Interferon-Driven Oxysterol-Based Defense against Tumor-Derived Extracellular Vesicles. *Cancer Cell* **35**, 33-45 e36 (2019).
10. Bhattacharya, S., *et al.* Inducible priming phosphorylation promotes ligand-independent degradation of the IFNAR1 chain of type I interferon receptor. *J Biol Chem* **285**, 2318-2325 (2010).
11. Bhattacharya, S., *et al.* Role of p38 protein kinase in the ligand-independent ubiquitination and down-regulation of the IFNAR1 chain of type I interferon receptor. *J Biol Chem* **286**, 22069-22076 (2011).
12. Katlinskaya, Y.V., *et al.* Type I Interferons Control Proliferation and Function of the Intestinal Epithelium. *Mol Cell Biol* **36**, 1124-1135 (2016).
13. Liu, J., *et al.* Mammalian casein kinase 1alpha and its leishmanial ortholog regulate stability of IFNAR1 and type I interferon signaling. *Mol Cell Biol* **29**, 6401-6412 (2009).
14. Zheng, H., Qian, J., Baker, D.P. & Fuchs, S.Y. Tyrosine phosphorylation of protein kinase D2 mediates ligand-inducible elimination of the Type 1 interferon receptor. *J Biol Chem* **286**, 35733-35741 (2011).
15. Zheng, H., Qian, J., Varghese, B., Baker, D.P. & Fuchs, S. Ligand-stimulated downregulation of the alpha interferon receptor: role of protein kinase D2. *Mol Cell Biol* **31**, 710-720 (2011).
16. Liu, J., *et al.* Virus-induced unfolded protein response attenuates antiviral defenses via phosphorylation-dependent degradation of the type I interferon receptor. *Cell Host Microbe* **5**, 72-83 (2009).
17. Liu, J., *et al.* Ligand-independent pathway that controls stability of interferon alpha receptor. *Biochem Biophys Res Commun* **367**, 388-393 (2008).
18. Bhattacharya, S., *et al.* Anti-tumorigenic effects of Type 1 interferon are subdued by integrated stress responses. *Oncogene* **32**, 4214-4221 (2013).
19. Qian, J., *et al.* Pathogen recognition receptor signaling accelerates phosphorylation-dependent degradation of IFNAR1. *PLoS Pathog* **7**, e1002065 (2011).
20. HuangFu, W.C., Liu, J., Harty, R.N. & Fuchs, S.Y. Cigarette smoking products suppress anti-viral effects of Type I interferon via phosphorylation-dependent downregulation of its receptor. *FEBS Lett* **582**, 3206-3210 (2008).

21. Huangfu, W.C., *et al.* Inflammatory signaling compromises cell responses to interferon alpha. *Oncogene* **31**, 161-172 (2012).
22. Bhattacharya, S., *et al.* Triggering ubiquitination of IFNAR1 protects tissues from inflammatory injury. *EMBO Mol Med* **6**, 384-397 (2014).
23. Katlinski, K.V., *et al.* Inactivation of Interferon Receptor Promotes the Establishment of Immune Privileged Tumor Microenvironment. *Cancer Cell* **31**, 194-207 (2017).
24. Carbone, C.J. & Fuchs, S.Y. Eliminative signaling by Janus kinases: role in the downregulation of associated receptors. *J Cell Biochem* **115**, 8-16 (2014).
25. Fuchs, S.Y. Hope and fear for interferon: the receptor-centric outlook on the future of interferon therapy. *J Interferon Cytokine Res* **33**, 211-225 (2013).
26. Zheng, H., *et al.* Vascular endothelial growth factor-induced elimination of the type 1 interferon receptor is required for efficient angiogenesis. *Blood* **118**, 4003-4006 (2011).
27. Cho, C., *et al.* Cancer-associated fibroblasts downregulate type I interferon receptor to stimulate intratumoral stromagenesis. *Oncogene* **39**, 6129-6137 (2020).
28. Crotta, S., *et al.* Type I and type III interferons drive redundant amplification loops to induce a transcriptional signature in influenza-infected airway epithelia. *PLoS Pathog* **9**, e1003773 (2013).
29. Davidson, S., Crotta, S., McCabe, T.M. & Wack, A. Pathogenic potential of interferon alphabeta in acute influenza infection. *Nat Commun* **5**, 3864 (2014).
30. Kohlmeier, J.E., Cookenham, T., Roberts, A.D., Miller, S.C. & Woodland, D.L. Type I interferons regulate cytolytic activity of memory CD8(+) T cells in the lung airways during respiratory virus challenge. *Immunity* **33**, 96-105 (2010).
31. Mordstein, M., *et al.* Interferon-lambda contributes to innate immunity of mice against influenza A virus but not against hepatotropic viruses. *PLoS Pathog* **4**, e1000151 (2008).
32. Pulverer, J.E., *et al.* Temporal and spatial resolution of type I and III interferon responses in vivo. *J Virol* **84**, 8626-8638 (2010).
33. Seo, S.U., *et al.* Type I interferon signaling regulates Ly6C(hi) monocytes and neutrophils during acute viral pneumonia in mice. *PLoS Pathog* **7**, e1001304 (2011).
34. de Weerd, N.A., *et al.* Structural basis of a unique interferon-beta signaling axis mediated via the receptor IFNAR1. *Nat Immunol* **14**, 901-907 (2013).
35. Shepardson, K.M., *et al.* IFNAR2 Is Required for Anti-influenza Immunity and Alters Susceptibility to Post-influenza Bacterial Superinfections. *Front Immunol* **9**, 2589 (2018).

REVIEWERS' COMMENTS

Reviewer #1 (Remarks to the Author):

Comments have been successfully addressed. One minor suggestion to make Figure 1D more clear is to include values from controls and spontaneous tumor models side by side (rather than the fold change)

Reviewer #2 (Remarks to the Author):

No more comments.

Reviewer #3 (Remarks to the Author):

The authors have addressed all major issues raised during the original review. This manuscript makes an important contribution to the field.

We thank our reviewers for their positive consideration of our work. The comments were as follows:

Reviewer #1 (Remarks to the Author):

Comments have been successfully addressed. One minor suggestion to make Figure 1D more clear is to include values from controls and spontaneous tumor models side by side (rather than the fold change)

- We thank our reviewer for positive assessment of our work. We appreciate the comment. The reason for presenting the data as fold of changes was because different pairs of transgenic mice and their respective control littermates were evaluated at different time (often months apart) due to limited breeding activity of mice. As a result we observed substantial variability of IFNAR1 expression between the experiments. Therefore to reduce this variability we assessed fold changes from control littermates in each separate experiment.

Reviewer #2 (Remarks to the Author):

No more comments.

Reviewer #3 (Remarks to the Author):

The authors have addressed all major issues raised during the original review. This manuscript makes an important contribution to the field.